# A systematic review and meta-analysis of the global prevalence and determinants of COVID-19 vaccine acceptance and uptake in people living with HIV

Sahabi Kabir Sulaiman [1] ✉, Muhammad Sale Musa [1], Fatimah Isma'il Tsiga-Ahmed [2], Abdulwahab Kabir Sulaiman [3,4] & Abdulaziz Tijjani Bako [5]

People living with HIV (PLHIV) are at higher risk of poor outcomes of SARS-CoV-2 infection. Here we report the pooled prevalence of COVID-19 vaccine acceptance/uptake and determinants among this vulnerable population of PLHIV based on a systematic review and meta-analysis of studies published by 25 August 2023. Among the 54 included studies ($N$ = 167,485 participants), 53 ($N$ = 166,455) provided data on vaccine acceptance rate, while 27 ($N$ = 150,926) provided uptake data. The global prevalences of COVID-19 vaccine acceptance and uptake were 67.0% and 56.6%, respectively. Acceptance and uptake rates were 86.6% and 90.1% for the European Region, 74.9% and 71.6% for the Region of the Americas, 62.3% and 78.9% for the South-East Asian Region, 64.6% and 19.3% for the Eastern Mediterranean Region, 58.0% and 35.5% for the African Region, and 57.4% and 44.0% for the Western Pacific Region. The acceptance rate increased from 65.9% in 2020 to 71.0% in 2022, and the uptake rate increased from 55.9% in 2021 to 58.1% in 2022. Men, PLHIV aged ≥40 years and those who had recently received the influenza vaccine were more likely to accept and receive the COVID-19 vaccine. Factors associated with lower uptake included Black race, other races (Latinx/Hispanic/mixed race), low education level and being unemployed. Vaccine-related factors associated with higher acceptance included belief in vaccine effectiveness, vaccine trust, perceived high susceptibility to SARS-CoV-2 infection and fear of potential COVID-19 effect in PLHIV. Sustained efforts and targeted interventions are needed to reduce regional disparities in COVID-19 vaccine uptake among PLHIV.

Vaccination against severe acute respiratory syndrome coronavirus 2 (SARS-CoV-2) has been identified as one of the most potent public health intervention tools for containing the COVID-19 pandemic[1–3]. Evidence has shown the effectiveness of the vaccines against SARS-CoV-2 (refs. 3–6), including the more lethal Delta (B.1.617.2) variant[7,8]. Moreover, a recent study showed that COVID-19 vaccines are more effective than natural immunity at preventing all-cause emergency department visits, hospitalizations and deaths[9]. Despite these cumulative benefits,

[1]Department of Medicine, Yobe State University Teaching Hospital, Damaturu, Nigeria. [2]Department of Community Medicine, Bayero University Kano/Aminu Kano Teaching Hospital, Kano, Nigeria. [3]Department of Medicine, Murtala Muhammad Specialist Hospital, Kano, Nigeria. [4]Kwanar Dawaki COVID-19 Isolation Center, Kano, Nigeria. [5]Department of Neurosurgery, Houston Methodist, Houston, TX, USA. ✉e-mail: sahabikabir25@gmail.com

**Fig. 1 | PRISMA flow diagram of the literature search.** From a total of 5,739 studies identified (following both the initial and updated literature review in the select databases), we screened 4,462 studies for eligibility, removed 1,263 duplicates, and excluded 4,422 studies (1) not reporting COVID-19 vaccine acceptance/uptake rates and/or determinants, (2) reporting only on conditional acceptance (including willingness to pay for vaccination), (3) not reporting on PLHIV, (4) that were COVID-19 vaccination clinical trials with no report on the proportion of COVID-19 vaccine acceptance and/or uptake and (5) that employed a continuous variable for evaluating COVID-19 vaccine acceptability. Therefore, we finally included a total of 54 eligible studies for this systematic literature review and meta-analysis.

the available evidence demonstrates a recent global rise in "vaccine hesitancy"[10], defined by the World Health Organization (WHO) as "a delay in the acceptance or refusal of vaccination despite the availability of vaccination services"[11] and ranked among the top ten major threats to global health[12]. This unprecedented rise in vaccine hesitancy witnessed during the ongoing COVID-19 pandemic is believed to be largely due to COVID-19-related 'infodemic'[13–15], a phenomenon defined by WHO as 'too much information including false or misleading information in digital and physical environments during a disease outbreak'[16].

Given their compromised immune function, people living with human immunodeficiency virus (PLHIV) (especially those with suboptimal viral suppression/low CD4 count) have a higher risk of contracting infectious diseases and experiencing severe outcomes following exposure than non-HIV-infected people[17,18]. Specifically, evidence has indicated that, compared with non-HIV-infected people, PLHIV have a higher risk of SARS-CoV-2 hospitalizations[19–21] and death[17,19–23]. The safety, efficacy and immunogenicity of the COVID-19 vaccines, which were developed to help mitigate the enormous global burden of morbidity and mortality associated with SARS-CoV-2, have been proven[4,24–26], and vaccines employing mRNA technologies (such as the BNT162b2 and the mRNA-1273 vaccines) have been shown to be superior in terms of efficacy[4,26]. The safety, efficacy and immunogenicity of the COVID-19 vaccines have been established even in patients with background immunosuppressive states[27,28], including PLHIV[29,30].

Despite this cumulative evidence, COVID-19 acceptance/hesitancy and uptake remained highly variable among PLHIV across different WHO regions and sociodemographic contexts of the world, with some evidence even suggesting a higher rate of COVID-19 vaccine hesitancy among PLHIV than among people not living with HIV[31,32]. Moreover, previous systematic reviews and meta-analytic studies of COVID-19

vaccine acceptability have been conducted globally[33–42], regionally[43–45], nationally[46–48] and in certain cohorts[49–53], including children[54,55]. However, none of these reviews specifically evaluated COVID-19 vaccine acceptability and uptake and their respective determinants among PLHIV, a population that is highly susceptible to and heavily affected by SARS-CoV-2 infection.

We therefore conducted this systematic review and meta-analysis to evaluate the prevalence rates and time trends of COVID-19 vaccine acceptance and uptake among PLHIV. We further evaluated the pooled vaccine uptake rates among PLHIV who indicated their willingness to accept the vaccine and assessed the factors associated with COVID-19 vaccine acceptance/uptake.

## Results

### Study identification and selection
Of the 3,256 studies screened, 54 studies qualified for inclusion and were analysed (Fig. 1). The characteristics of the included studies are provided in Supplementary Table 1.

### Characteristics of the studies
This report included 54 studies (cumulative sample size, 167,485 participants), with men constituting the majority of the participants in 30 of the included studies[32,56–84]. The smallest sample size among the included studies was 15 (ref. 85), and the largest was 101,205 (ref. 82). A majority of the included studies (33 studies) were started and completed in the year 2021, and only 4 studies were started and completed in 2020 (refs. 68,78,86,87). Furthermore, 7 studies began in 2020 but were completed in 2021 (refs. 63,71,73,76,85,88,89), while 5 studies began in 2021 but were completed in 2022 (refs. 79,82,90–92), and 5 studies were started and completed in 2022 (refs. 81,84,93–95). In terms of

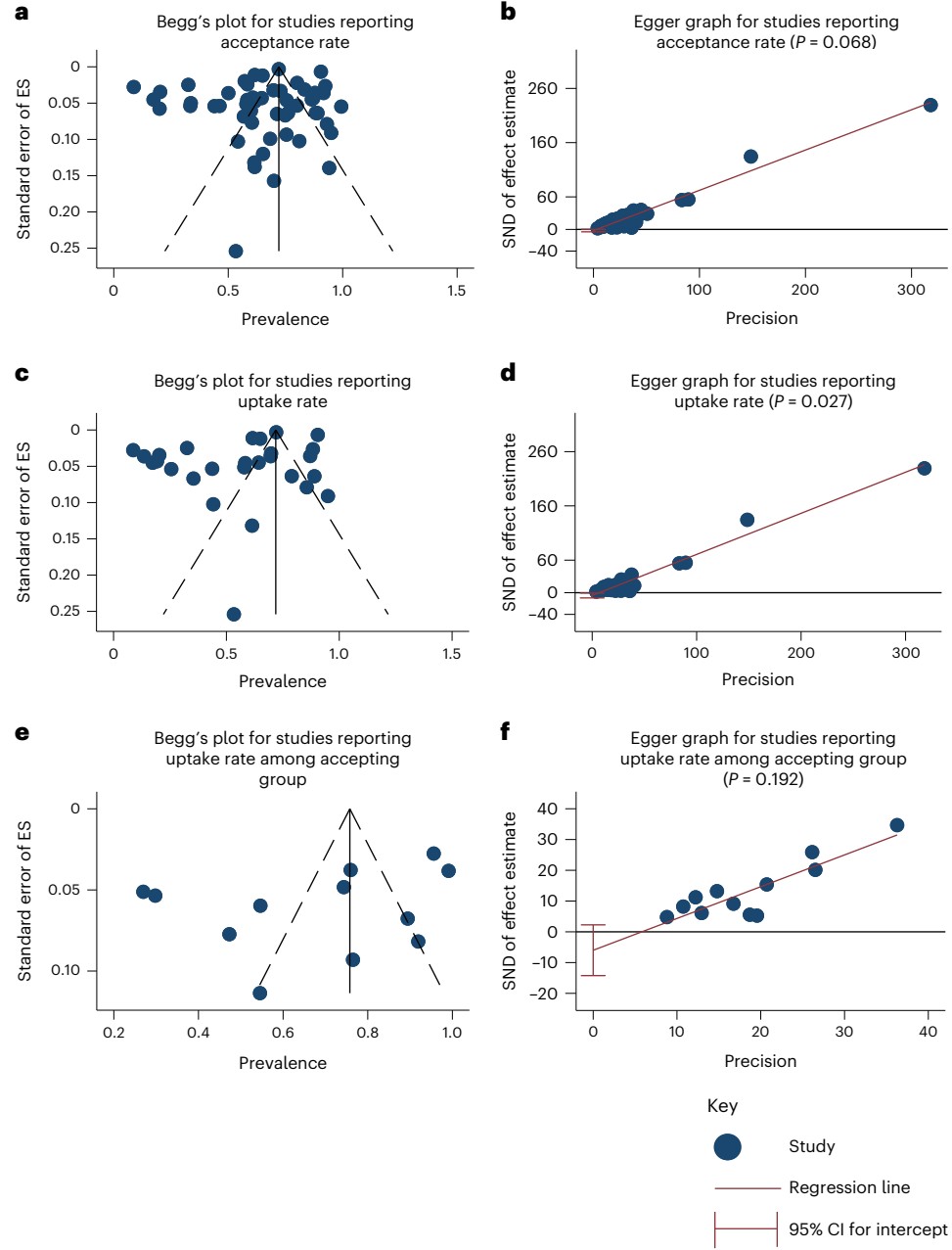

**Fig. 2 | Assessment of publication bias among studies reporting each of the three outcomes. a**, Begg's funnel plot of included studies reporting acceptance rate (*N* = 53 studies). **b**, Egger graph of included studies reporting acceptance rate (*P* = 0.068) (*N* = 53 studies). **c**, Begg's funnel plot of included studies reporting uptake rate (*N* = 27 studies). **d**, Egger graph of included studies reporting uptake rate (*P* = 0.027) (*N* = 27 studies). **e**, Begg's funnel plot of studies reporting uptake rate in PLHIV who indicated acceptance (*N* = 13 studies). **f**, Egger graph of studies reporting uptake among accepting group (*P* = 0.192) (*N* = 13 studies). All the statistical tests were two-sided. ES, effect size; SE, standard error; SND, standard normal distribution.

publication year, 2 studies were published in 2020 (refs. 68,86), 11 were published 2021 (refs. 32,57,58,60,62,64,66,68–70,88), 26 were published in 2022 (refs. 31,56,59,61,63,65,67,71–73,75–77,79,86,87,89,90,96–103) and 15 studies were published in 2023 (refs. 80–85,91–95,104–107).

The included studies spanned across all WHO regions, with 11 studies from the African Region (AFR) (South Africa, Uganda, Nigeria, Ethiopia, Zimbabwe, Sierra Leone and Malawi)[66,71,89,90,93–95,98,100,101,105], 18 from the Region of the Americas (AMR) (Canada, the United States and Latin America)[31,61,63,68,72,73,76,78,82,84–86,91,92,96,99,104,107], 6 from the European Region (EUR) (France, Ireland, Spain, and the UK)[60,64,79,88,97,106], 1 from the Eastern Mediterranean Region (EMR) (Egypt, Saudi Arabia and Tunisia)[74], 4 from the South-East Asian Region (SEAR) (India and Indonesia)[62,80,83,102] and 10 from the Western Pacific Region (WPR) (Australia and China)[32,56–58,65,67,69,70,75,77,81,103]. Two were multiregional studies (MRS) that involved PLHIV from multiple WHO regions[87,97]. By country, the United States had the highest number of studies (*N* = 14) (refs. 61, 63,68,73,76,78,82,85,86,91,92,96,99,107), followed by China (*N* = 11) (refs. 32,56–59,65,67,69,70,81,103), followed by Nigeria (*N* = 4) (refs. 90, 98,101,105). Stratified by study design, the majority (*N* = 45) of the studies were cross-sectional[31,32,56–60,63,65–81,83,84,86–90,93–96,98–106]. One study employed a retrospective medical charts review[64], and four studies did not report their sampling technique[61,62,97,107]. With regard to sampling method, 10 studies employed probability sampling techniques[31,56,66,67,71,83,97–99], 39 employed non-probability sampling

techniques[32,57–63,65,68–70,72–81,84–87,89,90,92–96,100–106] and 5 did not report their sampling technique[64,82,88,91,107]. A majority of the studies (*N* = 26) were conducted online[31,57–60,63,65,67,69,70,72–77,81,82,84,86,87,91,92,99,106,107]. However, 24 studies used face-to-face interviews[32,56,61,64,66,68,71,78–80,85,88–90,93–98,100,103–105], while 4 combined online and face-to-face interviews[83,96,101,102].

The acceptance rates for the COVID-19 vaccine among PLHIV were reported by 39 studies[31,56,58–60,62,64,66,67,70–73,75,76,78–81,83,86–88,90,92,93,95,96,98–105], while 26 studies reported the rates of vaccination (uptake)[32,56–58,61,65,74,75,77,79,81,82,84,85,89,91,92,94,96,97,99,100,102,105–107], although 1 study[63] reported neither intention to accept nor the actual uptake. Among studies that used a multivariable regression analysis, 19 reported factors associated with acceptance[31,56,59,60,66,68,70,71,73,74,76,83,95,96,98–100,103,104], 11 studies reported factors associated with uptake[61,65,69,81,84,91,97,99,100,104,107] and 7 reported factors associated with hesitancy[57,62,63,81,90,93,95].

### Risk of publication bias
No evidence of publication bias was observed by visual inspection of Begg's funnel plot (Fig. 2a) or by using Egger's test (*P* = 0.068) for studies evaluating acceptance (Fig. 2b). However, slight evidence of publication bias was observed among studies reporting uptake rates using both Begg's funnel plot (Fig. 2c) and Egger's test (*P* = 0.027) (Fig. 2d). No evidence of publication bias was observed by visual inspection of Begg's funnel plot (Fig. 2e) or by using Egger's test (*P* = 0.137) (Fig. 2f) for studies reporting uptake rates among PLHIV who indicated acceptance.

### Sensitivity analyses
The leave-one-out sensitivity analysis showed that the reported pooled rate of COVID-19 vaccine acceptance was not individually influenced by a single study, with the pooled rates varying between 65.0% (95% confidence interval (CI), 59.0–71.0%; *P* = 0.000) and 67.0% (95% CI, 60.1–72.0%; *P* = 0.000) across the sequential iterations of the leave-one-out analysis (Supplementary Fig. 1). Also, across all sequential iterations, we did not observe evidence of an overriding influence of a single study on the pooled uptake rate, with the pooled rates ranging from 54.0% (95% CI, 44.0–64.0%; *P* = 0.000) to 58.0% (95% CI, 48.0–68.0%; *P* = 0.000) across all iterations (Supplementary Fig. 2). Similarly, there was no evidence of an overriding influence of a single study on the pooled uptake rate among PLHIV who indicated acceptance, with the pooled rates varying between 52.0% (95% CI, 38.0–67.0%; *P* = 0.000) and 58.0% (95% CI, 45.0–72.0%; *P* = 0.000) across all the sequential iterations (Supplementary Fig. 3).

### Meta-analysis of the COVID-19 vaccine acceptance rates
The results of the meta-analysis of the prevalence of COVID-19 vaccine acceptance among PLHIV are shown in Fig. 3a. The pooled global acceptance rate of the COVID-19 vaccine among PLHIV from 53 studies, with a cumulative sample size of 166,455, was 67.0% (95% CI, 62.0–71.9%) (Supplementary Fig. 4). The prevalence of acceptance, however, increased from 65.9% (95% CI, 55.0–76.0%) in 2020 to 71.0% (95% CI, 57.2–832%) in 2022 (Supplementary Fig. 5).

Stratified by WHO region (Fig. 4a), EUR had the highest prevalence at 86.6% (95% CI, 73.5–95.8%), followed by AMR with 74.9% (95% CI, 68.0–81.2%), then EMR with 64.6% (95% CI, 60.5–68.5%), then SEAR with

62.3% (95% CI, 33.8–86.9%), AFR with 58.0% (95% CI, 40.3–74.7%) and WPR with 57.4% (95% CI, 41.2–72.9%) (Supplementary Fig. 6). The pooled acceptance rate of MRS was 65.4% (95% CI, 64.3–66.5%) (Supplementary

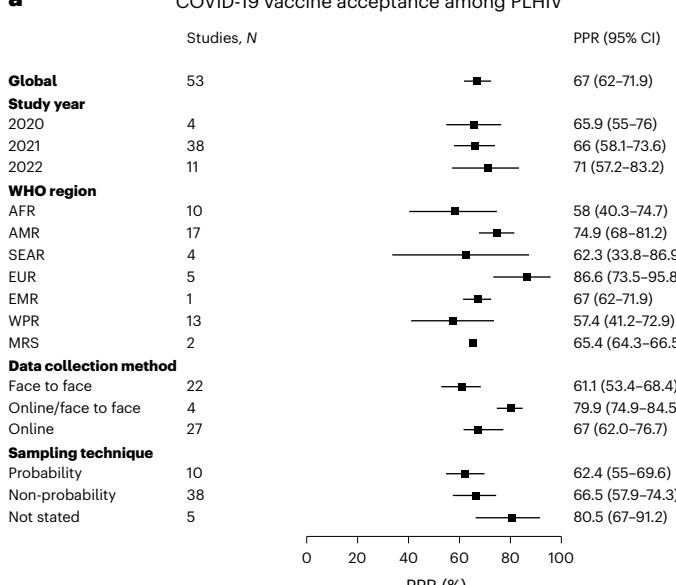

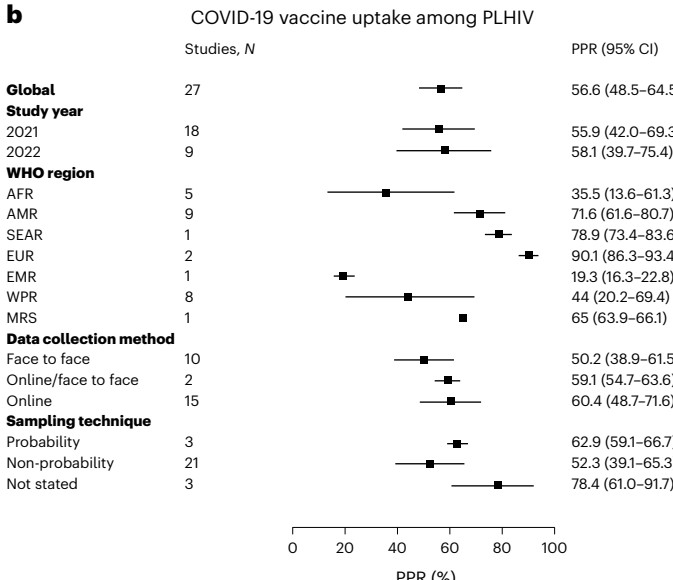

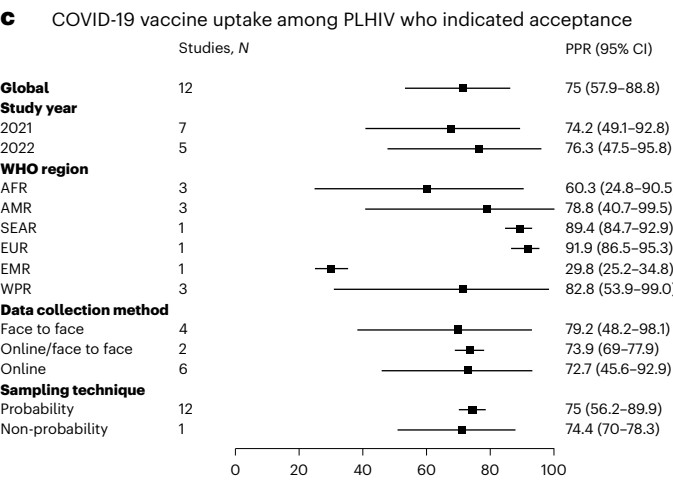

**Fig. 3 | Forest plots of the results of a random-effects model meta-analysis of the prevalence rates including subgroup analyses for the three proportion outcomes pooled using inverse variance weights. a**, Prevalence of COVID-19 vaccine acceptance among PLHIV (*N* = 53 studies, *N* = 166,455 participants). **b**, Prevalence of COVID-19 vaccine uptake among PLHIV (*N* = 27 studies, *N* = 150,926 participants). **c**, Prevalence of COVID-19 vaccine uptake among PLHIV who indicated acceptance (*N* = 13 studies, *N* = 6,564 participants). Each solid square represents the effect size of each characteristic, while the ends of the horizontal lines represent lower (left) and upper (right) CIs. All statistics were based on a two-sided *t*-test. PPR, pooled prevalence rate.

Fig. 6). Furthermore, the pooled acceptance rate among studies that used face-to-face interviews (61.1%; 95% CI, 53.4–68.4%), was lower than the pooled rate among studies that used online interviews (69.6%; 95% CI, 62.0–76.7%) (Supplementary Fig. 7). Similarly, the pooled acceptance rate among studies that used non-probability sampling techniques (66.4%; 95% CI, 57.9–74.3%) was lower than the rate among studies that used probability sampling techniques (62.4%; 95% CI, 55.0–69.6%) (Supplementary Fig. 8).

## Meta-analysis of the COVID-19 vaccine uptake rates

The results of the meta-analysis of the prevalence of COVID-19 vaccine uptake among PLHIV are shown in Fig. 3b. Overall, 27 studies, with a cumulative sample size of 150,926, reported the rate of vaccine uptake. The global uptake prevalence among these studies was 56.6% (95% CI, 48.5–64.5%) (Supplementary Fig. 9). Stratified by year of study, the pooled uptake prevalence increased from 55.9% (95% CI, 42.0–69.3%) in 2021 to 58.1% (95 CI, 39.7–75.4%) in 2022 (Supplementary Fig. 10).

Stratified by WHO region (Fig. 4b), uptake was highest in EUR (90.1%; 95% CI, 86.3–93.4%), followed by SEAR (78.9%; 95% CI, 73.4–83.6%), then AMR (71.6%; 95% CI, 61.6–80.7%), WPR (44.0%; 95% CI, 20.2–69.4%), AFR (35.5%; 95% CI, 13.6–61.3%) and EMR (19.3%; 95% CI, 16.2–22.8%) (Supplementary Fig. 11). The uptake prevalence from one MRS was 65.0% (95% CI, 63.9–66.1%) (Supplementary Fig. 11). Furthermore, uptake was substantially higher among studies employing online interviews (60.4%; 95% CI, 48.7–71.6%) than among studies using face-to-face interviews (50.2%; 95% CI, 38.9–61.5%) (Supplementary Fig. 12). The pooled uptake prevalence was also higher among studies that used probability sampling techniques (62.9%; 95% CI, 59.1–66.7%) than among those that used non-probability sampling (52.3%; 95% CI, 39.1–65.3%) (Supplementary Fig. 13).

## Meta-analysis of the COVID-19 vaccine uptake rates among PLHIV who indicated acceptance

The results of the meta-analysis of the prevalence of COVID-19 vaccine uptake among PLHIV who indicated acceptance are shown in Fig. 3c. The global vaccine uptake among PLHIV who indicated acceptance, which was reported in 13 studies with a cumulative sample size of 6,186, was 71.3% (95% CI, 53.3–86.4%) (Supplementary Fig. 14).

Stratified by WHO region (Fig. 4c), uptake among PLHIV who indicated acceptance was 91.9% (95% CI, 86.5–95.3%) in EUR, 89.4% (95% CI, 84.7–92.9%) in SEAR, 78.8% (95% CI, 40.7–99.5%) in AMR, 71.7% (95% CI, 31.3–98.1%) in WPR, 60.3% (95% CI, 24.8–90.5%) in AFR and 29.8% (95% CI, 25.2–34.8%) in EMR (Supplementary Fig. 15). Stratified by study year, uptake was 76.3% (95% CI, 47.5–95.8%) in 2022 and 67.6% (95% CI, 40.8–89.4%) in 2021 (Supplementary Fig. 16). The uptake reported among studies using face-to-face interviews (69.6%; 95% CI, 38.5–93.1%) was lower than the rate reported by online studies (72.7%; 95% CI, 45.6–92.9%) (Supplementary Fig. 17). Only one study reporting uptake rate used the probability sampling technique. The uptake rate from this study (74.4%; 95% CI, 70.0–78.3%) was slightly higher than the pooled uptake rate of studies that used non-probability sampling techniques (71.0%; 95% CI, 51.1–87.5%) (Supplementary Fig. 18). Supplementary Figs. 19–39 contain the meta-analysis results (peer-reviewed articles only) of all the three study outcomes (acceptance, uptake and uptake among PLHIV who indicated acceptance).

## Regional acceptance/uptake rates among PLHIV versus the general population

Table 1 shows a comparison of COVID-19 vaccine acceptance rates among PLHIV and the general population in each WHO region. There is evidence of substantial variability in acceptance rates across regions among both the general population (mean acceptance rate, 67.0; s.d., 8.4; range, 55.4–74.9%) and the PLHIV population (mean acceptance rate, 67.03; s.d., 11.4; range, 57.4–86.6%). However, the variability in regional uptake rates among PLHIV (mean uptake rate, 56.6; s.d., 27.7;

range, 19.3–90.1%) is substantially higher than that of the general population (mean acceptance rate, 40.6; s.d., 9.7; range, 25.7–52.0%) (Table 2).

## Meta-analysis of the factors associated with COVID-19 vaccine acceptance

Table 3 shows the results of individual meta-analyses of the factors associated with COVID-19 vaccine acceptance. Men had a higher likelihood of acceptance than women (odds ratio (OR), 2.06; 95% CI, 1.16–3.66). Conversely, PLHIV aged less than 40 years had significantly lower odds of acceptance than those aged 40 years and above (OR, 0.70; 95% CI, 0.54–0.90). Also, compared with those with secondary and higher levels of education, PLHIV having a primary level of education and below had a lower likelihood of acceptance (OR, 0.60; 95% CI, 0.40–0.89). Similarly, compared with PLHIV belonging to the white race, Black PLHIV had a significantly lower likelihood of acceptance (OR, 0.50; 95% CI, 0.27–0.94). However, compared with PLHIV who reside in urban settings, those who reside in rural settings had a significantly higher likelihood of acceptance (OR, 1.69; 96% CI, 1.33–2.14). Furthermore, PLHIV who are concerned about the safety of the COVID-19 vaccine had a lower likelihood of acceptance than those who are not (OR, 0.41; 95% CI, 0.32–0.53), whereas PLHIV who believe in the effectiveness of the COVID-19 vaccine (OR, 1.80; 95% CI, 1.27–2.56) and those who trust the vaccine (OR, 15.17; 95% CI, 9.16–25.12) had significantly higher odds of acceptance. Also, PLHIV who perceive that they are at an increased susceptibility of contracting COVID-19 (OR, 1.34; 95% CI, 1.07–1.68) and those who are fearful of the potential effect of COVID-19 on PLHIV (OR, 2.01; 95% CI, 1.60–2.54) had higher odds of acceptance. Furthermore, we found that PLHIV with a history of recent influenza vaccination uptake had a significantly higher likelihood of acceptance than those who have no recent influenza vaccine uptake history (OR, 2.01; 95% CI, 1.60–2.54). Supplementary Figs. 40–56 contain the outputs of the meta-analyses of the determinants of COVID-19 vaccine acceptance in PLHIV.

## Meta-analysis of the factors associated with COVID-19 vaccine uptake

Table 4 shows the results of meta-analyses of the factors associated with COVID-19 vaccine uptake. We found that men had significantly higher odds of vaccine uptake than women (OR, 1.55; 95% CI, 1.27–1.89). Also, compared with those aged 40 years and above, PLHIV aged below 40 years had significantly lower odds of vaccine uptake (OR, 0.58; 95% CI, 0.53–0.64). PLHIV who have attained at least a secondary level of education had a higher likelihood of uptake than those with only primary education or below (OR, 0.50; 95% CI, 0.41–0.61), and being unemployed was associated with a lower likelihood of vaccine uptake (OR, 0.56; 95% CI, 0.43–0.73). Furthermore, compared with PLHIV belonging to the white race, Black PLHIV (OR, 0.60; 95% CI, 0.52–0.70) as well as PLHIV belonging to other races (Latinx/Hispanic/mixed race) had a significantly lower likelihood of vaccine uptake (OR, 0.31; 95% CI, 0.28–0.34). Conversely, PLHIV who have recently been vaccinated against influenza had a significantly higher likelihood of COVID-19 vaccine uptake (OR, 6.73; 95% CI, 6.11–7.14). Supplementary Figs. 57–63 contain the results of the meta-analyses of the determinants of COVID-19 vaccine uptake among PLHIV.

## Discussion

This study evaluated the pooled prevalence and determinants of COVID-19 vaccine acceptance and uptake among the vulnerable population of PLHIV. The overall acceptance rate of the COVID-19 vaccine across the 53 studies reporting acceptance rates was approximately 67%. This aggregate acceptance rate varied when studies were stratified on the basis of WHO region, with the highest rate observed among PLHIV in EUR (86.6%), followed by AMR (74.9%), while the lowest acceptance rates were observed among PLHIV in AFR (58.0%) and WPR (57.4%). The aggregate uptake rate among studies reporting vaccination rates

**a**

Acceptance rates of the COVID-19 vaccine among PLHIV according to WHO region

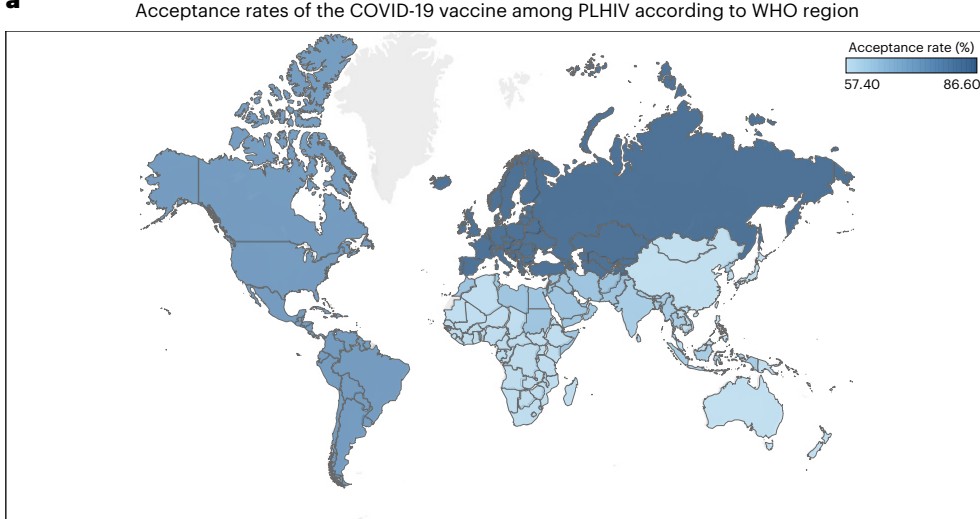

**b**

Uptake rates of the COVID-19 vaccine among PLHIV according to WHO region

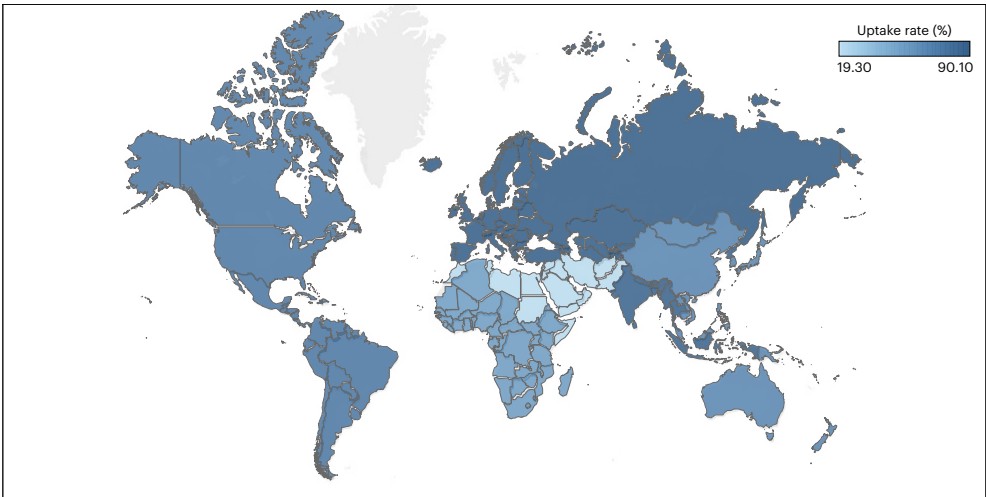

**c**  Uptake rates of the COVID-19 vaccine among PLHIV who indicated acceptance according to WHO region

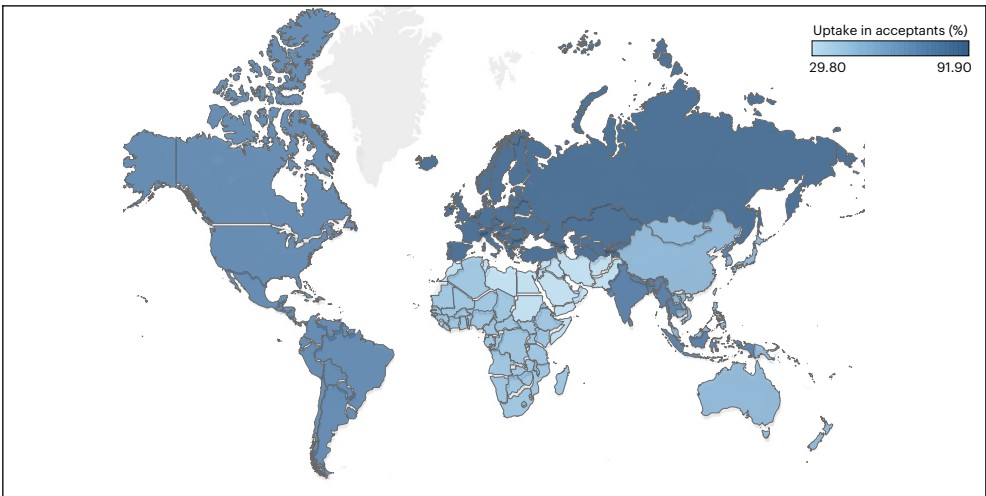

**Fig. 4 | Global map showing the regional pooled prevalence of COVID-19 vaccine acceptance/uptake. a**, Acceptance rates of the COVID-19 vaccine among PLHIV according to WHO global regions. **b**, Uptake rates of the COVID-19 vaccine among PLHIV according to WHO global regions. **c**, Uptake rates of the COVID-19 vaccine among PLHIV who indicated acceptance according to WHO global regions. Darker areas signify higher rates, and lighter areas signify lower rates. Maps adapted from OpenStreetMap under a Creative Commons licence CC BY-SA 2.0.

**Table 1 | COVID-19 vaccine acceptance rate of PLHIV across the six WHO regions compared to rates in the general population of the same region**

| WHO region | Study-derived regional acceptance rates of PLHIV (%) | Regional acceptance rates in the general population (based on re-analysis of a previous systematic review)[4] (%) |
|---|---|---|
| AFR | 58.0 (95% CI, 40.3–74.7) | 55.4 (95% CI, 51.1–59.7) |
| AMR | 74.9 (95% CI, 68.0–81.2) | 71.5 (95% CI, 69.8–73.2) |
| EMR | 64.6 (95% CI, 60.5–68.5) | 57.3 (95% CI, 53.4–61.3) |
| EUR | 86.6 (95% CI, 73.5–95.8) | 71.1 (95% CI, 67.7–74.4) |
| SEAR | 62.3 (95% CI, 33.8–86.9) | 74.9 (95% CI, 69.5–80.0) |
| WPR | 57.4 (95% CI, 41.2–72.9) | 72.0 (95% CI, 68.5–75.2) |

**Table 2 | COVID-19 vaccine uptake rate of PLHIV across the six WHO regions compared to the WHO-reported uptake rate among the general population of the same region**

| WHO region | Study-derived regional uptake rate among PLHIV (%) | Regional uptake rates in the general population (based on the results of a previous systematic review)[4] (%) |
|---|---|---|
| AFR | 35.5 (95% CI, 13.6–61.3) | 39.7 (95% CI, 22.4–58.4) |
| AMR | 71.6 (95% CI, 61.6–80.7) | 45.0 (95% CI, 37.0–53.1) |
| EMR | 19.3 (95% CI, 16.2–22.8) | 25.7 (95% CI, 14.9–38.3) |
| EUR | 90.1 (95% CI, 86.3–93.4) | 52.0 (95% CI, 41.2–62.7) |
| SEAR | 78.9 (95% CI, 73.4–83.6) | 47.4 (95% CI, 28.9–66.2) |
| WPR | 44.0 (95% CI, 20.2–69.4) | 33.7 (95% CI, 22.2–46.2) |

was approximately 56.6%, with variations across WHO regions: 90.1% in EUR, 71.6% in AMR, 35.5% in AFR and 44.0% in WPR.

The aggregate acceptance:uptake ratio for the COVID-19 vaccine observed among PLHIV in the present review (67.0%:56.6%) is comparable to that of the general population (67.8%:42.3%) (ref. 108). Only 71.3% of PLHIV who indicated their acceptance actually received at least a single dose of the COVID-19 vaccine, demonstrating the presence of a substantial gap between vaccine acceptance and uptake. Lack of access to the COVID-19 vaccine may hinder the ability of those who indicated their willingness to accept the vaccine to receive it, thereby increasing the gap between willingness to accept and actual uptake[109,110]. Future studies should evaluate the extent to which lack of vaccine access contributes to the observed acceptance–uptake gap. Also, because prior studies have demonstrated the existence of sociodemographic disparities in access to the COVID-19 vaccines[111–118], future studies should also investigate the extent to which sociodemographic differences in vaccine accessibility may contribute to differences in COVID-19 vaccine uptake rates among those who are already willing to accept the vaccine. Also, changes in individuals' attitudes towards vaccines (for example, due to recent exposure to misinformation) may explain the observed gap between acceptance and uptake rates. For example, one randomized controlled trial found that recent exposure to misinformation led to a drop in willingness to be vaccinated by more than 6% in both the United States and the United Kingdom among people who were initially willing to be vaccinated against COVID-19 (ref. 119). A more proactive escalation of vaccination campaign efforts may therefore be needed to reduce the wide gap between acceptance and actual uptake. Furthermore, gaps between vaccine acceptance and uptake rates have

been reported in other health-related interventions[120–125], particularly in relation to missed opportunity for vaccination[126]. Overall, research is needed to understand and quantify the magnitude of the root causes of non-uptake among those who have demonstrated their willingness to receive the vaccine.

This review also found substantial regional variability in the acceptance rates of the COVID-19 vaccine among PLHIV, with acceptance rates ranging from 57.4% in WPR to 86.6% in EUR. Notably, in WPR, the acceptance rate observed among the general population (72.0%) was substantially higher than the rate among PLHIV (57.4%), whereas in EUR, the acceptance rate among PLHIV (86.6%) was substantially higher than that of the general population (71.1%). However, in all other WHO regions, the COVID-19 vaccine acceptance rates among PLHIV and the general population were similar, suggesting that the inter-regional variability in vaccine acceptance rate observed among PLHIV is probably a function of acceptance variability among the general population. In contrast, our meta-analysis of uptake rates demonstrated evidence of substantially wider regional variability among PLHIV (mean uptake rate, 56.6; s.d., 27.7; range, 19.3–90.1%) than in the general population (mean acceptance rate, 40.6; s.d., 9.7; range, 25.7–52.0%). This observed variability in uptake rate may have been occasioned by the highly variable vaccination rollout among WHO member countries, with some countries such as the United Kingdom starting national vaccination programmes as early as December 2020, and others, especially some low-income and middle-income countries, commencing vaccination after April 2021[127,128]. Also, global inequities in access to and distribution of the COVID-19 vaccination may have substantially contributed to inter-regional variability in vaccine uptake rate. Nonetheless, the uptake rate of all six WHO regions was below WHO's target of 100% COVID-19 vaccination among immunocompromised populations such as PLHIV. Vaccination campaigns should therefore prioritize PLHIV, particularly in regions where the uptake rate is much lower than the general population's 70% target coverage[129].

Sociodemographic and regional variations in vaccination rates have been shown to play a leading role in the spread of new SARS-CoV-2 strains and the emergence of new waves across the globe[130–132]. Similar to previous studies[33–36,112,115–117], our study indicates that women (versus men), Black people and people of other races (Latinx/Hispanic/mixed race) (versus white), unemployed (versus employed) individuals, younger adults (<40 years versus ≥40 years), and those with a lower level of education are significantly less likely to receive at least one dose of the COVID-19 vaccine. These findings demonstrate that sociodemographic factors contribute to variations in vaccine uptake rates, signifying the need for policymakers to identify and address the sociodemographic determinants of uptake of the COVID-19 vaccine. Among other measures, vaccine confidence campaigns targeting sociodemographic subgroups with a lower likelihood of uptake and ensuring equitable access to and distribution of COVID-19 vaccines may substantially improve vaccine uptake among PLHIV.

The findings from our subgroup analyses indicate that the pooled acceptance rate in 2022 (71.0%) was higher than the rates in 2021 (66.0%) and 2020 (65.9%), and the uptake rate in 2022 (58.1%) was higher than the uptake rate in 2021 (55.9%). Similarly, among PLHIV who indicated acceptance, the uptake was higher in 2022 than in 2021 (76.3% versus 67.6%). This rising trend in the rate of acceptance of the COVID-19 vaccine over time was also observed in previous global studies, where the acceptance rate rose from 71.5% (ref. 133) in 2020 to 75.2% (ref. 134) in 2021, and to 79.1% (ref. 135) in 2022. This implies that the improvement in the acceptance of the COVID-19 vaccine over time among PLHIV is commensurate with that of the general population. Also, this increase in COVID-19 vaccine acceptance over time may reflect a positive change in the public's attitude towards vaccination due to deliberate vaccination campaign efforts or a gradual increase in access to COVID-19 vaccination globally[127,128].

**Table 3 | Results of meta-analyses of determinants of COVID-19 vaccine acceptance**

| Outcome | No. of studies and references | WHO region(s) represented | OR (95% CI) | P | I² within |
|---|---|---|---|---|---|
| Gender (men versus women) | 16, refs. 58–60,62,66,70–73,90,93,95,98,99,104,105 | AFR, AMR, EUR, SEAR, WPR | **2.06 (1.16–3.66)** | **0.01** | 95% |
| Age (<40 years versus ≥40 years) | 9, refs. 32,58,60,70–72,90,93,98 | AFR, AMR, EUR, WPR | **0.70 (0.54–0.90)** | **0.006** | 70% |
| Marital status (single/divorced/widowed versus married/cohabited) | 13, refs. 32,58,59,66,70,73,81,90,93,95,98,104,105 | AFR, AMR, WPR | 0.96 (0.80–1.15) | 0.83 | 38% |
| Race (Black versus white) | 2, refs. 73,99 | AMR | **0.50 (0.27–0.94)** | **0.03** | 75% |
| Race (others (Latinx/Hispanic/mixed race) versus white) | 2, refs. 73,99 | AMR | 1.04 (0.49–2.20) | 0.92 | 0% |
| Education (primary and below versus secondary and above) | 14, refs. 32,56,58,59,62,66,70,81,90,93,95,98,99,104 | AFR, AMR, SEAR, WPR | **0.60 (0.40–0.89)** | **0.01** | 93% |
| Employment status (unemployed versus employed) | 11, refs. 32,66,70,73,90,93,95,98,99,104,105 | AFR, AMR, WPR | 0.96 (0.66–1.31) | 0.20 | 77% |
| Income (low versus medium/high) | 9, refs. 58,59,66,70,73,81,90,93,98 | AFR, AMR, WPR | 0.96 (0.74–1.24) | 0.74 | 75% |
| Residence (rural versus urban) | 4, refs. 72,90,93,95 | AFR, AMR | **1.69 (1.33–2.14)** | **<0.001** | 94% |
| Comorbidity (absent versus present) | 9, refs. 58–60,66,70,72,81,98,104 | AFR, AMR, EUR, WPR | 0.81 (0.49–1.34) | 0.41 | 91% |
| Vaccine safety concern (yes versus no) | 4, refs. 56,60,62,98 | AFR, EUR, SEAR, WPR | **0.41 (0.32–0.53)** | **<0.001** | 96% |
| Perceived vaccine effectiveness (yes versus no) | 3, refs. 56,60,98 | AFR, EUR, WPR | **1.80 (1.27–2.56)** | **0.001** | 93% |
| Vaccine trust (yes versus no) | 3, refs. 60,62,104 | AMR, EUR, SEAR | **15.17 (9.16–25.12)** | **<0.001** | 31% |
| Perceived susceptibility to COVID-19 (yes versus no) | 3, refs. 59,60,90 | AFR, EUR, WPR | **1.34 (1.07–1.68)** | **0.01** | 0% |
| Fear of COVID-19 effect on PLHIV (yes versus no) | 4, refs. 59,60,71,98 | AFR, EUR, WPR | **2.01 (1.60–2.54)** | **<0.001** | 89% |
| Know someone who died of COVID-19 (yes versus no) | 2, refs. 60,62 | EUR, SEAR | 1.06 (0.68–1.66) | 0.78 | 82% |
| Recent of influenza vaccination (yes versus no) | 4, refs. 60,70–72 | AFR, AMR, EUR, WPR | **1.53 (1.29–1.81)** | **<0.001** | 36% |

Bold font indicates statistical significance.

**Table 4 | Results of meta-analyses of determinants of COVID-19 vaccine uptake**

| Outcome | No. of studies and references | WHO region(s) represented | OR (95% CI) | P | I² within |
|---|---|---|---|---|---|
| Gender (men versus women) | 6, refs. 58,61,64,69,84,99 | AMR, EUR, WPR | **1.55 (1.27–1.89)** | **<0.001** | 33% |
| Age (<40 years versus ≥40 years) | 4, refs. 58,61,69,84 | AMR, WPR | **0.58 (0.53–0.64)** | **<0.001** | 96% |
| Marriage/cohabitation (single/divorced/widowed versus married/cohabited) | 2, refs. 58,69 | WPR | 0.86 (0.68–1.09) | 0.21 | 68% |
| Race (Black versus white) | 3, refs. 61,84,99 | AMR | **0.60 (0.52–0.70)** | **<0.001** | 66% |
| Race (others (Latinx/Hispanic/mixed race) versus white) | 2, refs. 61,99 | AMR | **0.31 (0.28–0.34)** | **<0.001** | 91% |
| Education (primary and below versus secondary and above) | 5, refs. 58,69,81,84,99 | AMR, WPR | **0.50 (0.41–0.61)** | **<0.001** | 73% |
| Income (low versus medium/high) | 2, refs. 58,84 | AMR, WPR | 0.91 (0.61–1.34) | 0.62 | 34% |
| Employment (unemployed versus employed) | 4, refs. 64,69,84,99 | AMR, EUR, WPR | **0.56 (0.43–0.73)** | **0.002** | 74% |
| Receipt of influenza vaccination (yes versus no) | 2, refs. 61,69 | AMR, WPR | **6.73 (6.11–7.14)** | **<0.001** | 93% |

Bold font indicates statistical significance.

Finally, similar to previous reviews among certain immunocompromised groups[136,137] and other vaccines[120,138–142], this review found that participants' vaccine-related perceptions and attitudes (including scepticism about vaccine efficacy, concern about vaccine safety, prior influenza vaccination history, perceived heightened susceptibility to SARS-CoV-2 infection and fear of contracting the infection) are significantly predictive of COVID-19 vaccine acceptability and uptake. We want to note that not all PLHIV are the same. It is reasonable to expect

that PLHIV on antiretroviral therapy with an undetectable viral load, compared with those with uncontrolled HIV, may be more likely to have concerns about the safety of the COVID-19 vaccine among PLHIV. Future studies are therefore needed to evaluate the extent to which inter-regional disparities in access to antiretroviral therapy contribute to regional variations in vaccine acceptance and uptake among PLHIV. Nonetheless, interventions aimed at maximizing COVID-19 vaccine acceptance and uptake among PLHIV should prioritize health education about the proven safety and efficacy of the COVID-19 vaccines among PLHIV and individuals with other immune-compromising conditions[4,24,26,29,30,143,144]. Furthermore, interventions should address the root causes of poor attitudes towards the COVID-19 vaccine, including the ongoing unprecedented proliferation of misleading and false information in the digital and physical media, termed an "infodemic"[13] by WHO[16]. Among other measures, promoting personal health behaviours[145,146], maintaining the HIV care continuum[147–150] and incentivizing the uptake of the COVID-19 vaccine among PLHIV (especially among those with socio-economically disadvantaged backgrounds) may help bolster vaccination and combat hesitancy.

### Strengths and limitations

This systematic review has some strengths and limitations. First, we report the pooled estimate of the rate of uptake of the COVID-19 vaccine specifically among those who indicated acceptance, not just the overall uptake rate. Second, approximately 90% of the included studies scored high in terms of methodological rigor, and up to one fourth of the included studies employed a probability sampling technique. Third, we performed a series of stratified analyses to account for differences in sampling and data collection methods, yet our estimates of the prevalence of vaccine uptake and acceptance remained relatively similar. Fifth, when we excluded the two non-peer-reviewed[94,96] articles from our analyses, publication bias remained absent, and the prevalence rates of all the analyses remained similar (Supplementary Figs. 19–39). The main limitations of this review relate to those of the included studies. First, the cross-sectional nature of most of the included studies precludes our ability to establish causal relationships. Also, many of the studies used an online medium for questionnaire administration rather than face-to-face interviews, which may have resulted in the non-participation of PLHIV without access to the internet, a group that may account for a high proportion of the population of PLHIV, particularly those in Africa.

### Conclusion

There is substantial regional variation in the rates of acceptance and uptake of the COVID-19 vaccine among PLHIV, and approximately one third of PLHIV who were willing to accept the vaccine were yet to be vaccinated. Low levels of education, unemployment, and poor perception of and attitude towards the COVID-19 vaccines are among the main predictors of lower acceptance and undervaccination among PLHIV. Interventions at the global, national and local levels should therefore seek to address these barriers in order to improve acceptance and uptake of the COVID-19 vaccine among PLHIV.

## Methods

This review was performed in accordance with the Preferred Reporting Items for Systematic Reviews and Meta-analyses (PRISMA)[151] guidelines. The PRISMA checklist of this study is contained in Supplementary Table 2. The study protocol was registered with the International Prospective Register for Systematic Review (PROSPERO ID: CRD42022353575).

Between 10 and 15 January 2023, we searched multiple electronic databases, including PubMed, Scopus, Cochrane Database of Systematic Reviews, Cochrane Central Register of Controlled Trials, APA PsycInfo, CINAHL and Google Scholar, to identify studies assessing COVID-19 vaccine acceptance, uptake and hesitancy among PLHIV. We further updated our literature search to 25 August 2023 (following editorial

recommendation). A detailed search strategy was developed for Pub-Med and adapted for the other databases (Supplementary Table 3). We used a combination of Boolean operators ('AND' and 'OR'), Medical Subject Headings, key terms and wildcards to expand the search. The search terminologies used included 'coronavirus', 'COVID-19', 'SARS-CoV-2', 'vaccine', 'vaccination', 'HIV', 'AIDS', 'acceptance', 'willingness', 'intention', 'uptake', 'hesitancy', 'refusal', 'determinants', 'associated factors' and 'predictors'.

### Inclusion and exclusion criteria

For the prevalence of acceptance/uptake, we followed the CoCoPop (condition, context, population)[152] guideline for the review of prevalence/incidence studies and included any study that evaluated and reported the rates of COVID-19 vaccine acceptance, uptake and/or hesitancy among PLHIV. To evaluate the factors associated with acceptance/uptake, we included any study that provided information on the predictors of COVID-19 vaccine acceptance and/or uptake among PLHIV in line with the PEO (population, exposure, outcome)[152] guideline. We only included original full-text articles, preprints and abstracts evaluating any of our outcomes of interest (prevalence rates or factors associated with COVID-19 vaccine acceptability in PLHIV) that were reported in the English language. Preprints were included because evidence has shown that they are reliable in health decision-making during disease outbreaks, including the COVID-19 pandemic[153,154].

Records were excluded if they (1) did not provide information about any of our primary outcomes of interest, (2) evaluated and provided information on conditional acceptance only, such as willingness to pay for the vaccine; (3) employed only continuous variables for measuring acceptability; or (4) were COVID-19 vaccination clinical trials with no report of the proportions of the COVID-19 vaccine acceptability.

### Study selection and eligibility

Following a literature search in the select databases, 1,993 articles were screened. Using the Rayyan QCRI (Qatar Computing Research Institute)[155], the studies were independently screened on the basis of title, then abstract and full text by two investigators (S.K.S. and M.S.M.), and two senior authors (F.I.T.-A. and A.T.B.) resolved all discrepancies. Of the 1,993 studies screened, 167 were found to be eligible, and 40 studies were included in the analysis on the basis of our pre-specified inclusion and exclusion criteria (Fig. 1). In addition, we searched the databases between January and 25 August 2023 for relevant studies published within this period, and 2,483 articles were retrieved, of which 14 were eligible and included to update our initial search (Fig. 1). The total number of studies included in this review after this update was therefore 54 (Fig. 1).

### Outcomes and definitions

The primary outcomes of this systematic review and meta-analysis are the prevalence of COVID-19 vaccine acceptance and vaccine uptake among PLHIV. For this meta-analysis, we defined acceptance (similar to a previous meta-analysis)[108] as the willingness to be vaccinated against COVID-19 (among unvaccinated people). Therefore, participants who responded 'Yes', 'Definitely' or 'Probably' to the question evaluating their willingness or intention to accept the COVID-19 vaccine were considered to be in the vaccine acceptance group. Vaccine uptake was defined as the receipt of at least a single dose of the COVID-19 vaccine. For articles that reported only the prevalence of uptake, we considered the reported uptake rate as the acceptance rate, in line with a previously published study[108]. Prevalence of acceptance was calculated as the number of participants in the acceptance group divided by the total number of participants multiplied by 100. Accordingly, the prevalence of vaccine uptake was calculated as the number of participants in the uptake group divided by the total number of participants multiplied by 100. We further calculated the prevalence of vaccine uptake among those who indicated willingness to accept the COVID-19 vaccine as

the number of participants who received the vaccine divided by the number of participants who indicated willingness to accept the vaccine, multiplied by 100.

## Data extraction
The included articles from the literature search were initially entered into Zotero software (version 6.0.15), where duplicates were detected and removed. Subsequently, the Joanna Briggs Institute[156] data extraction form was used to extract all relevant data from all included articles by two investigators (S.K.S. and M.S.M.). The information extracted from the articles included the first author's name, the year of publication, the study country, the study setting, the study design, the sample size, the sampling method, the study period, the participants' demographic characteristics (age and sex), the total proportion of those willing to be vaccinated, the total proportion of those who were already vaccinated and the factors associated with vaccine acceptance. Additionally, for each subcategory of each determinant of vaccine acceptance and uptake (for example, male versus female), we extracted the total number of participants in each subcategory and the number of participants in each subcategory who indicated vaccine acceptance and/or uptake. At least two studies had to provide such data before a determinant was considered for meta-analysis. All extracted data are openly available at https://doi.org/10.17605/OSF.IO/5XGJE.

## Critical appraisal (quality assessment) of the included studies
All included studies were independently reviewed by two investigators (S.K.S. and M.S.M.), who critically appraised the studies' methodological quality using the Joanna Briggs Institute[157] critical appraisal tool. This checklist assesses the methodological rigor of a study using nine unique questions about the study's sample frame, participants, sample size, study setting, data analysis, methods for identifying conditions, measurement procedures, statistical analysis and adequate response rate. A study was graded in terms of quality on the basis of the overall score, with greater than 70% indicating high quality, 50% to 70% indicating moderate quality and less than 50% indicating low quality[36,158]. Differences in scoring between the two investigators (S.K.S. and M.S.M.) were resolved by two senior authors (F.I.T.-A. and A.T.B.) by reviewing and discussing the articles together before finally awarding a consensus score (Supplementary Tables 4 and 5).

## Statistical analysis
The aggregate rates of COVID-19 vaccine acceptance and uptake were estimated from all studies with meta-analysis weighting. Studies that reported both willingness to accept (among unvaccinated people) and the actual uptake of COVID-19 vaccines were used to estimate the uptake rates among PLHIV who indicated acceptance. All pooled proportions were presented using forest plots. A random-effects model was chosen due to the anticipated diversity of the study populations and the variability in the timing of studies. The percentage of total variation across studies due to heterogeneity was evaluated using the $I^2$ measure[159,160], which was categorized as low (0% to 25%), moderate (26% to 75%) or substantial (76% to 100%).

Subgroup analysis was performed on the basis of WHO region (AFR, AMR, EMR, EUR, SEAR and WPR), year of study, data collection method, sampling method and method for questionnaire administration. In the subgroup analysis by region, studies involving participants from more than one WHO region with no exclusive outcome data per country were treated independently as MRS. The Freeman–Tukey double arcsine transformation was enabled to prevent the exclusion of some studies with proportions close to or at 1. The pooled proportions and weighted mean differences with their 95% CIs were presented, and a $P$ value of 0.05 was considered significant. To examine the potential influence of each study on the pooled rates of COVID-19 vaccine acceptance and uptake, we performed a leave-one-out sensitivity analysis, which involved an iterative exclusion of one study from the analysis

to report the pooled estimates of vaccine acceptance/uptake without that study. This process was repeated until all included studies had been individually excluded.

The presence of publication bias among the included studies was checked using Begg's funnel plot[161] and Egger's test[162]. A value of $P > 0.05$ indicates the absence of statistically significant evidence of publication bias. All meta-analyses of prevalence rates and publication bias assessments were performed using the metaprop[163] command in Stata version 15IC (StataCorp).

Furthermore, we performed a meta-analysis of the potential determinants of acceptance and uptake using RevMan-5 software (version 5.4.1, Nordic Cochrane Centre) where at least two studies had provided the required data for the meta-analysis of a particular determinant. Using the Mantel–Haenszel method, we calculated the pooled effect estimates (ORs) of each determinant and their 95% CIs using either random-effects or fixed-effects meta-analyses, depending on the degree of heterogeneity across the studies[164,165], and presented the results as forest plots. To derive the pooled estimate of vaccine acceptance/uptake among the general population of each WHO region, we conducted meta-analyses using data from a recently published global meta-analysis of COVID-19 vaccine acceptance and uptake in the general population[108]. Finally, we compared the mean and standard deviation of the regional acceptance/uptake rates among PLHIV to those of the general population using an $F$-test.

## Reporting summary
Further information on research design is available in the Nature Portfolio Reporting Summary linked to this article.

## Data availability
All the data supporting the findings of this work can be accessed via the Open Science Framework at https://doi.org/10.17605/OSF.IO/5XGJE.

## Code availability
All the code for the data analysed in this work is openly available at https://doi.org/10.17605/OSF.IO/5XGJE.

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

## Acknowledgements

The authors received no specific funding for this work.

## Author contributions

Conceptualization (evolution of the original study idea, aims and goals): S.K.S. Methodology (design of the research methods including choice of literature search engines, search strategy, study selection and assessment of study quality): S.K.S., M.S.M., F.I.T.-A., A.K.S. and A.T.B. Formal analysis (publication bias assessment, sensitivity analysis and meta-analysis of prevalence rates and determinants): S.K.S., MSM

and A.T.B. Data curation (verification of the accuracy of extracted data, arrangement and coding of the data in analysis software and storage of the data for reuse): S.K.S., and M.S.M. Writing—original draft (preparation and presentation of the initial manuscript draft): S.K.S. Writing—review and editing (critical revisions and commentaries on the manuscript before and after peer review): A.T.B., F.I.T.-A., A.K.S., M.S.M. and S.K.S. Supervision (responsibility of leading the research process to completion): F.I.T.-A., A.K.S. and A.T.B. Validation (verification that the study results and outputs are reproducible): S.K.S., M.S.M., F.I.T.-A., A.K.S. and A.T.B. Investigation (extraction of all data from the included studies): SKS and MS.;Visualization (presentation of the publishable version): A.T.B. Project administration (coordination and management of all study processes from conception to execution): S.K.S.

## Competing interests

The authors declare no competing interests.

## Additional information

**Correspondence and requests for materials** should be addressed to Sahabi Kabir Sulaiman.

# Reporting Summary

## Statistics

For all statistical analyses, confirm that the following items are present in the figure legend, table legend, main text, or Methods section.

| n/a | Confirmed | |
|---|---|---|
| ☐ | ☒ | The exact sample size (*n*) for each experimental group/condition, given as a discrete number and unit of measurement |
| ☐ | ☒ | A statement on whether measurements were taken from distinct samples or whether the same sample was measured repeatedly |
| ☐ | ☒ | The statistical test(s) used AND whether they are one- or two-sided *Only common tests should be described solely by name; describe more complex techniques in the Methods section.* |
| ☐ | ☒ | A description of all covariates tested |
| ☐ | ☒ | A description of any assumptions or corrections, such as tests of normality and adjustment for multiple comparisons |
| ☐ | ☒ | A full description of the statistical parameters including central tendency (e.g. means) or other basic estimates (e.g. regression coefficient) AND variation (e.g. standard deviation) or associated estimates of uncertainty (e.g. confidence intervals) |
| ☐ | ☒ | For null hypothesis testing, the test statistic (e.g. *F*, *t*, *r*) with confidence intervals, effect sizes, degrees of freedom and *P* value noted *Give P values as exact values whenever suitable.* |
| ☒ | ☐ | For Bayesian analysis, information on the choice of priors and Markov chain Monte Carlo settings |
| ☒ | ☐ | For hierarchical and complex designs, identification of the appropriate level for tests and full reporting of outcomes |
| ☐ | ☒ | Estimates of effect sizes (e.g. Cohen's *d*, Pearson's *r*), indicating how they were calculated |

*Our web collection on statistics for biologists contains articles on many of the points above.*

## Software and code

Policy information about availability of computer code

| Data collection | Zotero software (version 6.0.15). The codes are available in the Supporting Information file and and the Open Source Framework page: https://osf.io/536qx/. |
|---|---|
| Data analysis | Stata Version 15IC (StataCorp, College Station, Texas USA), RevMan-5 Software (Version 5.4.1, The Nordic Cochrane Centre, Copenhagen) and Microsoft Excel and PowerPoint were used for data analysis and presentation. The direct outputs from these softwares are available in the Supporting Information file and and the Open Source Framework page: https://osf.io/536qx/. |

For manuscripts utilizing custom algorithms or software that are central to the research but not yet described in published literature, software must be made available to editors and reviewers. We strongly encourage code deposition in a community repository (e.g. GitHub). See the Nature Portfolio guidelines for submitting code & software for further information.

## Data

Policy information about availability of data

All manuscripts must include a data availability statement. This statement should provide the following information, where applicable:

- Accession codes, unique identifiers, or web links for publicly available datasets
- A description of any restrictions on data availability
- For clinical datasets or third party data, please ensure that the statement adheres to our policy

All data data supporting the findings of work presented here can be accessed via the Open Science Framewrok link, https://osf.io/536qx/.

# Human research participants

Policy information about studies involving human research participants and Sex and Gender in Research.

| | |
|---|---|
| Reporting on sex and gender | Carefully, we reported the findings based on gender as appropriate based on what was extracted from the included studies. |
| Population characteristics | See above |
| Recruitment | This was a systematic review and meta-analysis of available literature on the research subject at hand. |
| Ethics oversight | The study was pre-registered in the International Prospective Register for Systematic Review and Meta-analysis (PROSPERO), and the study has adhere to the Preferred Reporting Items For Systematic Review and Meta-analysis statement. |

Note that full information on the approval of the study protocol must also be provided in the manuscript.

# Field-specific reporting

Please select the one below that is the best fit for your research. If you are not sure, read the appropriate sections before making your selection.

☐ Life sciences     ☒ Behavioural & social sciences     ☐ Ecological, evolutionary & environmental sciences

For a reference copy of the document with all sections, see nature.com/documents/nr-reporting-summary-flat.pdf

# Behavioural & social sciences study design

All studies must disclose on these points even when the disclosure is negative.

| | |
|---|---|
| Study description | This was a systematic review and meta-analyses of quantitative studies reported any of the outcomes of interest. |
| Research sample | Global population of adult people living with HIV as reported in original studies done online/onsite in any place in the world. |
| Sampling strategy | No limitation regarding sampling strategy in our inclusion criteria so long as the study fulfilled all the inclusion criiteria |
| Data collection | Two researches (SKS and MSM) used the Joanna Briggs Institute data extraction form to extract relevant data from a;; included studies. |
| Timing | All studies published until 25th August 2023 were eligible for inclusion. |
| Data exclusions | 154 We only included original full-text articles, preprints, and abstracts evaluating any of our outcomes of interest (prevalence rates or factors associated with COVID-19 vaccine acceptability in PLHIV) that were reported in the English language |
| Non-participation | Not applicable. |
| Randomization | Not applicable |

# Reporting for specific materials, systems and methods

We require information from authors about some types of materials, experimental systems and methods used in many studies. Here, indicate whether each material, system or method listed is relevant to your study. If you are not sure if a list item applies to your research, read the appropriate section before selecting a response.

## Materials & experimental systems

| n/a | Involved in the study |
|---|---|
| ☒ | ☐ Antibodies |
| ☒ | ☐ Eukaryotic cell lines |
| ☒ | ☐ Palaeontology and archaeology |
| ☒ | ☐ Animals and other organisms |
| ☒ | ☐ Clinical data |
| ☒ | ☐ Dual use research of concern |

## Methods

| n/a | Involved in the study |
|---|---|
| ☒ | ☐ ChIP-seq |
| ☒ | ☐ Flow cytometry |
| ☒ | ☐ MRI-based neuroimaging |

