## [Peer Review File · Nature Human Behaviour]

Peer Review Information

Journal: Nature Human Behaviour

Manuscript Title: A systematic review and meta-analyses of the global prevalence and determinants of COVID-19 vaccine acceptance and uptake in people living with HIV

Corresponding author name(s): Sahabi Kabir Sulaiman

Reviewer Comments & Decisions:

Decision Letter, initial version:

15th May 2023

Dear Dr Sulaiman,

Thank you once again for your manuscript, entitled "The global prevalence and determinants of COVID-19 vaccine acceptance and uptake among people living with HIV: a systematic review and meta-analysis", and for your patience during the peer review process.

Your Article has now been evaluated by 2 referees. You will see from their comments copied below that, although they find your work of potential interest, they have raised quite substantial concerns. In light of these comments, we cannot accept the manuscript for publication, but would be interested in considering a revised version if you are willing and able to fully address reviewer and editorial concerns.

We hope you will find the referees' comments useful as you decide how to proceed. If you wish to submit a substantially revised manuscript, please bear in mind that we will be reluctant to approach the referees again in the absence of major revisions. We are committed to providing a fair and constructive peer-review process. Do not hesitate to contact us if there are specific requests from the reviewers that you believe are technically impossible or unlikely to yield a meaningful outcome.

In your revisions please:

1) Without comparison to vaccine acceptance & uptake rates in the general populations in question, the conclusions are unsupported. Please perform these comparisons.

2) Please include full (meta)analyses of determinants of uptake/acceptance as they are not as rigorously reported in the review as those on rates of acceptance and uptake. Please clearly indicate how many studies these results are based on.

3) Please perform a sensitivity analysis to investigate how each individual study included affects the overall pooled effect.

If you wish to submit a suitably revised manuscript, we would hope to receive it within 1.5 months. I would be grateful if you could contact us as soon as possible if you foresee difficulties with meeting this target resubmission date.

- Include a “Response to the editors and reviewers” document detailing, point-by-point, how you addressed each editor and referee comment. If no action was taken to address a point, you must provide a compelling argument. When formatting this document, please respond to each reviewer comment individually, including the full text of the reviewer comment verbatim followed by your response to the individual point. This response will be used by the editors to evaluate your revision and sent back to the reviewers along with the revised manuscript.
- Highlight all changes made to your manuscript or provide us with a version that tracks changes.

[REDACTED]

Thank you for the opportunity to review your work. Please do not hesitate to contact me if you have any questions or would like to discuss the required revisions further.

Sincerely,

Arunas Radzvilavicius, PhD
Senior Editor, Nature Human Behaviour
Nature Research

Reviewer expertise:

Reviewer #1: public health

Reviewer #2: public health, vaccination

REVIEWER COMMENTS:

Reviewer #1:

Remarks to the Author:

The manuscript "The global prevalence and determinants of COVID-19 vaccine acceptance and uptake among people living with HIV: a systematic review and meta-analysis" is a very well written and carefully performed review. However, I have a few comments/suggestions below:

Reviewer Comments

Question 1: In accordance with the PICO criteria please explain the inclusion criteria of the participants in the study.

Question 2: Have you conducted a sensitivity analysis to investigate how each individual study included affects the overall pooled effect? Could you please provide the results of the analysis?

Reviewer #2:

Remarks to the Author:

This is an ambitious review of studies that identify rates of acceptance and uptake of COVID-19 vaccines among persons living with HIV globally. The search strategy and criteria seem largely appropriate, and the review process well implemented. However, there are several fundamental challenges in the authors' analyses and interpretations that are not based on the studies they reviewed, and which make the overall review less rigorous and less interesting.

First, it would be very helpful to report the rates of vaccine acceptance and at least uptake among the general populations in each region as one wonders if the reported rates and their variability among PLHIV simply mirror that of the general population in their respective regions. As there seems to be little variability once these are reported in the Discussion, it would seem reasonable to assume that the regional differences in rates among PLHIV are merely a function of other regional differences that may have less to do with individuals' HIV status than other structural and community factors. This undercuts to an extent some of the authors' recommendations for specific information tailored for PLHIV as a means to address regional disparities.

Second, and relatedly, the authors seem to ascribe all gaps between reported vaccine acceptance and uptake to "hesitancy". This omits documented disparities in ACCESS to Covid-19 vaccines which have been found to vary by race/ethnicity and socioeconomic status in the US, and of course by region. To reflexively ascribe all these apparent deficiencies in uptake to personal decision-making and misinformation is mistaken, and arguably would fail to meaningfully effect changes in uptake.

Third, the authors' approach to reporting on determinants of acceptance and uptake needs to be better articulated. The previous rigor applied to defining and estimating acceptance and uptake is not similarly executed regarding determinants. For one, the correlates are often reported based on 1 study. It would seem erroneous to place much credence in one finding from one unidentified subpopulation in one unidentified region and to then generalize to others. There are also, for example, many articles that would not have met inclusion criteria for the present review, which report in depth on challenges around Covid-19 vaccine uptake among Black/African Americans in the US. As a result, most of the correlates presented are not very meaningful. Those that are corroborated across a number of studies might better be designated as such and differentiated from single-study results.

Additionally, studies in many areas of health care and technologies show tremendous gaps between stated acceptability and uptake. This should at least be noted among the study limitations.

A smaller point, but it would behoove the authors to present at least some context to indicate that not all PLHIV—even if reduced to their HIV status—are the same. The review itself might not be able to parse these issues, but one would expect there to be differences in susceptibility to and outcomes of Covid-19 among PLHIV who are on ART regimens with undetectable viral load vs. those who are not on ART or for whom it is not as effective. This is important as the results need to be interpreted more specifically rather than applied to PLHIV as a monolith. Arguably, one whose viral load is not controlled may have more reality-based concerns about vaccination than one with an immune system in the 'normal' range. And one wonders about regional differences in access to ART, usage of ART, and whether these may in part help to explain some of the reported disparities in uptake?

Finally, in one statement at the end of a paragraph (line 417), the authors note the importance of addressing gaps in SDOH across populations. However, 90% of their speculative analysis discounts SDOH and rather ascribes differences in uptake to cognitive and psychological factors (knowledge, attitudes, misinformation), omitting mention of tremendous global inequities in Covid-19 vaccine availability and access, in addition to those within regions and countries.

Author Rebuttal to Initial comments

REFEREE 1

Comment (1)

In accordance with the PICO criteria please explain the inclusion criteria of the participants in the study.

Response

We thank the Reviewer for this feedback. Because our study involves a review of prevalence rates and determinants, we have determined that the CoCoPop (for prevalence) and the PEO (for determinants) guidelines will better align with the peculiarities of our study. Therefore, we have made relevant changes to our Inclusion Criteria in line with these guidelines in the Methods section as follows:

For the prevalence of acceptance/uptake, we followed the CoCoPop (Condition, Context, Population)²⁴ guideline for the review of prevalence/incidence studies and included any study that evaluated and reported the rates of COVID-19 vaccine acceptance, uptake, and/or hesitancy among PLHIV. To evaluate the determinants of acceptance/uptake, we included any study that provided information on the determinants of COVID-19 vaccine acceptance and/or uptake among PLHIV in line with the PEO (Population, Exposure,

Outcome)²⁴ We only included original full-text articles, preprints, and abstracts evaluating any of our outcomes of interest (prevalence rates or determinants of COVID-19 vaccine acceptability in PLHIV) that were reported in the English language. Preprints were included because evidence has shown that they are reliable in health decision-making during disease outbreaks, including the COVID-19 pandemic.^{25,26} Records were excluded if they (1) did not provide information about any of our primary outcomes of interest; (2) if they only evaluated and provided information on conditional acceptance such as willingness to pay for the vaccine; (3) if they employed only continuous variables for measuring acceptability; or (4) if they were COVID-19 vaccination clinical trials with no report of the proportions of the COVID-19 vaccine acceptability.

Comment (2)

Have you conducted a sensitivity analysis to investigate how each individual study included affects the overall pooled effect? Could you please provide the results of the analysis?

Response

We thank the Reviewer for raising this important point. We've performed the sensitivity analyses for both outcomes among all included studies, as well as among the peer-reviewed studies, and added the figures in supplementary files (2 and 3), and made relevant changes within the main manuscript as follows:

Methods:

To examine the potential influence of each study on the pooled rates of COVID-19 vaccine acceptance and uptake, we performed a “leave-one-out” sensitivity analysis, which involved an iterative exclusion of one study from the analysis to report the pooled estimates of vaccine acceptance/uptake without that study and repeating this process until all included studies have been individually excluded.

Results:

Sensitivity analyses. The “leave-one-out” sensitivity analysis showed that the reported pooled rate of COVID-19 vaccine acceptance was not individually influenced by a single study, with the pooled rates varying between 63.0% (95% CI:57.0%-70.0%; $p = 0.000$) and 66.0% (95% CI:60.0%-71.0%; $p = 0.000$) across the sequential iterations of the “leave-one-out analysis” (**Figure 3 in Supplementary File 2**). Similarly, no evidence of an overriding influence of a single study was observed for pooled uptake rate, with the pooled rates varying between 45.0% (95% CI:34.0%-55.0%; $p = 0.000$) and 49.0% (95% CI:39.0%-60.0%; $p = 0.000$) across all sequential iterations of the “leave-one-out analysis” (**Figure 4 in Supplementary File 2**).

Random-effects REML model

Figure 3: Sensitivity analyses for each study reporting the acceptance rate of the COVID-19 vaccine among PLHIV

Random-effects REML model
Sorted by: _meta_id

Figure 4: Sensitivity analyses for each study reporting the uptake rate of the COVID-19 vaccine among PLHIV

REFEREE 2

This is an ambitious review of studies that identify rates of acceptance and uptake of COVID-19 vaccines among persons living with HIV globally. The search strategy and criteria seem largely appropriate, and the review process is well implemented. However, there are several fundamental challenges in the authors' analyses and interpretations that are not based on the studies they reviewed, and which make the overall review less rigorous and less interesting.

Response

We appreciate these valuable remarks made by the reviewer.

Comments (1)

First, it would be very helpful to report the rates of vaccine acceptance and at least uptake among the general populations in each region as one wonders if the reported rates and their variability among PLHIV simply mirror that of the general population in their respective regions. As there seems to be little variability once these are reported in the Discussion, it would seem reasonable to assume that the regional differences in rates among PLHIV are merely a function of other regional differences that may have less to do with individuals' HIV status than other structural and community factors. This undercuts to an extent some of the authors' recommendations for specific information tailored for PLHIV as a means to address regional disparities.

Response

We thank the reviewer for this important suggestion. We have, accordingly, performed these comparisons and updated all relevant

sections including our methods results and discussion sections as follows:

Methods:

To derive the pooled estimate of vaccine acceptance/uptake among the general population of each WHO region, we conducted meta-analyses, using data from a recently published global meta-analysis of COVID-19 vaccine acceptance and uptake among the general population.¹ Accordingly, we report a comparison of the mean, standard deviation, and range of the regional acceptance/uptake rates among PLHIV to those of the general population.

Results:

Comparison of the regional rates of acceptance and uptake of the COVID-19 vaccine among PLHIV with those of the general population in the respective WHO regions

Table 2 shows the rates of acceptance of the COVID-19 vaccine in each WHO region among PLHIV compared to the rates among the general population. There is evidence of substantial variability in acceptance rates across regions among both the general population (mean acceptance rate [SD] = 67.0 [8.4]; range = 55.4% - 74.9%) and the PLHIV population (mean acceptance rate [SD] = 67.2 [11.1]; range = 53.2% - 83.8%). However, in contrast to the variability of regional acceptance rates, the variability in regional uptake rates among PLHIV (mean uptake rate [SD] = 52.6 [25.6]; range = 19.3% - 85.6%) is substantially higher than that of the general population

(mean acceptance rate [SD] = 40.6 [9.7]; range = 25.7% - 52.0%),
 (Table 3).

WHO region	Study-derived regional acceptance rates of PLHIV (%)	Region acceptance rates in the general population (based on the results of a previous systematic review)¹
AFR	57.5% (95% CI:40.5%-73.6%)	55.4% (95% CI:51.1-59.7%)
AMR	71.8% (95% CI:62.7%-80.1%)	71.5% (95% CI:69.8-73.2%)
EMR	64.6% (95% CI:60.5%-68.5%)	57.3% (95% CI:53.4-61.3%)
EUR	83.8% (95% CI:67.9%-95.2%)	71.1% (95% CI:67.7-74.4%)
SEAR	72.4% (95% CI:69.0%-75.7%)	74.9% (95% CI:69.5-80.0%)
WPR	53.2% (95% CI:38.3%-67.7%)	72.0% (95% CI:68.5-75.2%)

Table 3: COVID-19 vaccine uptake rate of PLHIV across the six WHO regions compared to the WHO-reported uptake rate among the general population of the same region

WHO region	Study-derived regional uptake rate among PLHIV (%)	Regional uptake rates in the general population (based on the results of a previous systematic review)¹
AFR	40.6%(95% CI:14.1%-70.5%)	39.7% (95% CI:22.4%-58.4%)

AMR	54.3% (95% CI:41.9%-66.3%)	45.0% (95% CI:37.0%-53.1%)
EMR	19.3% (95% CI:16.2%-22.8%)	25.7% (95% CI:14.9%-38.3%)
EUR	85.6% (95% CI: 79.4%-90.2%)	52.0% (95% CI: 41.2%-62.7%)
SEAR	78.9% (95% CI:73.4%-83.6%)	47.4% (95% CI:28.9%-66.2%)
WPR	36.7% (95% CI:22.6%-52.0%)	33.7% (95% CI:22.2%-46.2%)

Discussion:

This review also found substantial regional variability in the acceptance rates of the COVID-19 vaccine among PLHIV, with acceptance rates ranging from 53.2% in the WPR to 83.8% in the EUR. Notably, in the WPR, the acceptance rate observed among the general population (72.0%) was substantially higher than the rate among PLHIV (53.2%). However, in all other WHO regions, the COVID-19 vaccine acceptance rate among PLHIV and the general population were closely similar, suggesting that the interregional variability in vaccine acceptance rate observed among PLHIV is essentially a function of acceptance variability among the general population. In contrast, our meta-analysis of uptake rates

demonstrated evidence of a substantially wider regional variability among PLHIV (mean uptake rate [SD] = 52.6 [25.6]; range = 19.3% - 85.6%), relative to the general population (mean acceptance rate [SD] = 40.6 [9.7]; range = 25.7% - 52.0%). This observed variability in uptake rate may have been occasioned by the highly variable vaccination rollout among WHO member countries, with some countries like the United Kingdom starting national vaccination program as early as December 2020, and others, especially some LMICs, commencing vaccination after April 2021.^{2,3} Also, global inequities in access to and distribution of the COVID-19 vaccination may have significantly contributed to inter-regional variability in vaccine uptake rate. Nonetheless, the uptake rate of all 6 WHO regions was below the WHO's target of 100% COVID-19 vaccination among immunocompromised populations like PLHIV. Therefore, vaccination campaigns, particularly in regions where the uptake rate is much lower than the general population's 70% target coverage, should prioritize PLHIV.⁴

Conclusion:

There is a substantial regional variation in the rate of acceptance and uptake of the COVID-19 vaccine among PLHIV, and approximately four-tenth of PLHIV who were willing to accept the vaccine were yet to be vaccinated. Low levels of education, unemployment, as well as the poor perception of and attitude toward the COVID-19 vaccines are among the main determinants of lower acceptance and under-vaccination among PLHIV. Therefore, interventions at the global, national, and local levels should seek to address these barriers

in order to improve acceptance and uptake of the COVID-19 vaccine among PLHIV.

Comment (2)

Second, and relatedly, the authors seem to ascribe all gaps between reported vaccine acceptance and uptake to “hesitancy”. This omits documented disparities in ACCESS to Covid-19 vaccines which have been found to vary by race/ethnicity and socioeconomic status in the US, and of course by region. To reflexively ascribe all these apparent deficiencies in uptake to personal decision-making and misinformation is mistaken and arguably would fail to meaningfully effect changes in uptake.

Response

We thank the Reviewer for this point. We’ve updated our Discussion section to capture the concept of “access to the COVID-19 vaccine” as follows:

Discussion: Lack of access to the COVID-19 vaccine may hinder the ability of those who indicated their willingness to accept the vaccine to receive it, thereby increasing the gap between willingness to accept and actual uptake.^{27,28} Future studies should evaluate the extent to which lack of vaccine access contributes to the observed acceptance-uptake gap. Also, because prior studies have demonstrated the existence of sociodemographic disparities in access to the COVID-19 vaccines,^{29–36} future studies should also investigate the extent to which sociodemographic differences in vaccine accessibility

may contribute to differences in COVID-19 vaccine uptake rates among those who are already willing to accept the vaccine.

Discussion: This observed variability in uptake rate may have been occasioned by the highly variable vaccination rollout among WHO member countries, with some countries like the United Kingdom starting national vaccination programs as early as December 2020, and others, especially some LMICs, commencing vaccination after April 2021.^{2,3} Also, global inequities in access to and distribution of the COVID-19 vaccination may have significantly contributed to inter-regional variability in vaccine uptake rate

Comment (3)

Third, the authors' approach to reporting on determinants of acceptance and uptake needs to be better articulated. The previous rigor applied to defining and estimating acceptance and uptake is not similarly executed regarding determinants. For one, the correlates are often reported based on 1 study. It would seem erroneous to place much credence in one finding from one unidentified subpopulation in one unidentified region and to then generalize it to others. There are also, for example, many articles that would not have met inclusion criteria for the present review, which report in depth on challenges around Covid-19 vaccine uptake among Black/African Americans in the US. As a result, most of the correlates presented are not very meaningful. Those that are corroborated across a number of studies might better be designated as such and differentiated from single-study results.

Response

We are grateful to the Reviewer for this very important suggestion. We have returned to all the 40 studies included and extracted relevant data to evaluate the pooled estimates of the potential effects of the determinants and performed two series of meta-analyses, for acceptance, and for uptake. Accordingly, we've also made changes to the relevant aspects in the Results, Methods, and Discussion aspects as follows:

Table 4: Results of meta-analyses of determinants of COVID-19 vaccine acceptance

Outcome	Studies (N) and references	WHO region(s) represented	OR (95% CI)	p-Value	I² Within
Gender (male vs female)	14 ⁵⁻¹⁸	AFR, AMR, EUR, SEAR, WPR	2.29 (1.19 - 4.39)	0.01	95%
Age (< 40 years vs >= 40 years)	8 ^{7-9,11,14,15,17,19}	AFR, AMR, EUR, WPR	0.70 (0.52 - 0.93)	0.01	70%
Marital status (single/divorced/widowed vs married/cohabited)	10 ^{5,8,9,11-13,15,16,18,19}	AFR, AMR, WPR	1.02 (0.86 - 1.21)	0.83	38%
Race (black vs others)	2 ^{5,10}	AMR	0.75 (0.41 - 1.35)	0.34	75%
Education (primary and below vs	11 ^{6,8-13,15,18-20}	AFR, AMR, SEAR, WPR	0.69 (0.43 - 1.10)	0.12	93%

secondary and above)					
Employment status (unemployed vs employed)	10 ^{5,8-13,15,16,19}	AFR, AMR, WPR	0.80 (0.57 - 1.13)	0.20	77%
Income (low vs medium/high)	7 ^{5,8,9,11,13,15,18}	AFR, AMR, WPR	0.96 (0.69 - 1.35)	0.83	75%
Comorbidity (absent vs present)	8 ^{8,9,12-15,17,18}	AFR, AMR, EUR, WPR	0.72 (0.42 - 1.25)	0.24	91%
Vaccine safety concern (yes vs no)	4 ^{6,9,17,20}	AFR, EUR, SEAR, WPR	0.41 (0.32 - 0.53)	<0.00001	96%
Perceived vaccine effectiveness (yes vs no)	3 ^{9,17,20}	AFR, EUR, WPR	1.80 (1.27 - 2.56)	0.001	93%
Vaccine trust (yes vs no)	3 ^{6,12,17}	AMR, EUR, SEAR	15.17 (9.16 - 25.12)	<0.00001	31%
Perceived susceptibility to COVID-19 (yes vs no)	3 ^{11,17,18}	AFR, EUR, WPR	1.34 (1.07 - 1.68)	0.01	0%
Fear of COVID-19 effect on PLHIV (yes vs no)	4 ^{7,9,17,18}	AFR, EUR, WPR	2.01 (1.60 - 2.54)	<0.00001	89%
Know someone who died of COVID-19	2 ^{6,17}	EUR, SEAR	1.06 (0.68 - 1.66)	0.78	82%

(Yes vs no)					
Recent of influenza vaccination (yes vs no)	4 ^{7,8,14,17}	AFR, AMR, EUR, WPR	1.53 (1.29 - 1.81)	<0.00001	36%

Results:

Meta-analysis of the determinants of COVID-19 vaccine acceptance.

Table 4 shows the results of individual meta-analyses of each determinant of COVID-19 vaccine acceptance. Males (vs. females) have a higher likelihood of acceptance (OR:2.29; 95% CI: 1.19 - 4.39). Conversely, PLHIV aged less than 40 years had significantly lower odds of acceptance compared to those aged 40 years and above (OR:0.70; 95% CI: 0.52 - 0.93). Furthermore, PLHIV who are concerned about the safety of the COVID-19 vaccine had a lower likelihood of acceptance compared to those who are not (OR: 0.41; 95% CI: 0.32 - 0.53). On the other hand, PLHIV who believe in the effectiveness of the COVID-19 vaccine (OR:1.80; 95% CI: 1.27 - 2.56), and those who trust the vaccine (OR:15.17; 95% CI: 9.16 - 25.12) had significantly higher odds of acceptance. Also, PLHIV who perceive that they are at an increased susceptibility of contracting COVID-19 (OR: 1.34; 95%: 1.07 - 1.68), and those who are fearful of the potential effect of COVID-19 on PLHIV (OR: 2.01; 95% CI: 1.60 - 2.54) have higher odds of acceptance. Furthermore, we found that PLHIV with a history of recent influenza vaccination uptake had a significantly higher likelihood of acceptance compared to those who

have no recent influenza vaccine uptake history (OR: 2.01; 95% CI: 1.60 - 2.54). **Supplementary File 4** contains the outputs of the meta-analysis of COVID-19 vaccine acceptance in PLHIV.

Table 5: Results of meta-analyses of determinants of COVID-19 vaccine uptake

Outcome	Studies (N) and references	WHO region(s) represented	OR (95% CI)	p-Value	I² Within
Gender (male vs female)	5 ^{10,15,21-23}	AMR, EUR, WPR	1.56 (1.38 - 1.77)	<0.00001	33%
Age (<40 years vs ≥ 40 years)	3 ^{15,21,23}	AMR, WPR	0.59 (0.54 - 0.65)	<0.00001	96%
Marriage/cohabitation (single/divorced/widowed vs married/cohabited)	2 ^{15,23}	WPR	0.86 (0.68 - 1.09)	0.21	68%
Race (Black vs others)	3 ^{10,21,22}	AMR, EUR	0.67 (0.57 - 0.77)	<0.00001	0%
Education (primary and below vs secondary and above)	3 ^{10,15,23}	AMR, WPR	0.49 (0.38 - 0.62)	<0.00001	73%
Employment	3 ^{10,22,23}	AMR, EUR,	0.63 (0.47 - 0.84)	0.002	74%

(unemployed vs unemployed)		WPR			
Receipt of influenza vaccination (yes vs no)	2 ^{21,23}	AMR, WPR	6.73 (6.11 - 7.14)	<0.00001	93%

Bold font within the p-Value column indicates statistical significance

Results:

Meta-analysis of the determinants of COVID-19 vaccine uptake.

Table 5 shows the results of meta-analyses of the determinants of COVID-19 vaccine uptake. We found that males had significantly higher odds of vaccine uptake, compared to females (OR:1.56; 95% CI: 1.38 - 1.77). Also, compared to those aged 40 years and above, PLHIV aged below 40 years had significantly lower odds of vaccine uptake (OR:0.59; 95% CI: 0.54 - 0.65). Furthermore, PLHIV who have attained at least a secondary level of education had a higher likelihood of uptake compared to those with primary or below (OR:0.49; 95% CI: 0.38 - 0.62), and being unemployed was associated with a lower likelihood of vaccine uptake (OR:0.63; 95% CI: 0.47 - 0.84). Also, compared to PLHIV belonging to other races, Black PLHIV have significantly lower odds of vaccine uptake (OR:0.67; 95% CI: 0.57 - 0.77). **Supplementary File 5** contains the outputs of the meta-analyses of COVID-19 vaccine uptake among PLHIV.

Additionally, studies in many areas of health care and technologies show tremendous gaps between stated acceptability and uptake. This should at least be noted among the study limitations.

Response

We thank the reviewer for this comment. We've added a note on this in the relevant aspect of the manuscript as follows:

Discussion:

Furthermore, gaps between vaccine acceptance and uptake rates have been reported in other health-related interventions,^{37–42} particularly in relation to a missed opportunity for vaccination (MOV)⁴³. Overall, research is needed to understand and quantify the magnitude of the root causes of non-uptake among those who have demonstrated their willingness to receive the vaccine.

Comment (5)

A smaller point, but it would behoove the authors to present at least some context to indicate that not all PLHIV—even if reduced to their HIV status—are the same. The review itself might not be able to parse these issues, but one would expect there to be differences in susceptibility to and outcomes of Covid-19 among PLHIV who are on ART regimens with undetectable viral load vs. those who are not on ART or for whom it is not as effective. This is important as the results need to be interpreted more specifically rather than applied to PLHIV as a monolith. Arguably, one whose viral load is not controlled may have more reality-based concerns about vaccination than one with an immune system in the 'normal' range. And one wonders about regional differences in access to ART, usage of ART, and whether these may in part help to explain some of the reported disparities in uptake?

Response

We thank the reviewer for this. We've mentioned this in our Introduction and now added a comment in the Discussion as follows:

Introduction: Given their compromised immune function, PLHIV (especially those with suboptimal viral suppression/low CD4 count) have an accelerated risk of contracting infectious diseases and experiencing severe outcomes following exposure, compared to non-HIV-infected people.^{44,45}

Discussion:

We want to note that not all PLHIV are the same. It is reasonable to expect that PLHIV on ART with an undetectable viral load, compared to those with uncontrolled HIV, may be more likely to have concerns about the safety of the COVID-19 vaccine among PLHIV. Therefore, future studies are needed to evaluate the extent to which inter-regional disparities in access to ART contribute to regional variations in vaccine acceptance and uptake among PLHIV. Nonetheless, interventions aimed at maximizing the COVID-19 vaccine acceptance and uptake among PLHIV should prioritize health education about the proven safety and efficacy of the COVID-19 vaccines among PLHIV and individuals with other immune-compromising

Comment (6)

Finally, in one statement at the end of a paragraph (line 417), the authors note the importance of addressing gaps in SDOH across populations. However, 90% of their speculative analysis

discounts SDOH and rather ascribe differences in uptake to cognitive and psychological factors (knowledge, attitudes, misinformation), omitting mention of tremendous global inequities in Covid-19 vaccine availability and access, in addition to those within regions and countries.

Response

We thank the reviewer for this important observation. We have substantially reviewed the discussion section and further emphasized the potential contribution of SDOH and disparities in access to vaccines on the observed regional variations in vaccine uptake rates.

Discussion:

Sociodemographic and regional variations in vaccination rates have been shown to play a leading role in the spread of new SARS-CoV-2 strains and the emergence of new waves across the globe.^{46–48} Similar to previous studies,^{30,33–35,49–52} our study indicates that females (vs. males), blacks (vs. others), unemployed (vs. employed) individuals, younger adults (<40 years vs. ≥40 years), and those with a lower level of education are significantly less likely to receive at least one dose of the COVID-19 vaccine. These findings demonstrate that sociodemographic factors may contribute to variations in vaccine uptake rates, signifying the need for policymakers to identify and address the sociodemographic determinants of uptake of the COVID-19 vaccine. Among other measures, vaccine confidence campaigns targeting sociodemographic subgroups with a lower likelihood of uptake, as well as ensuring equitable access to and distribution of COVID-19 vaccines may significantly improve vaccine uptake among PLHIV.

References

1. Wang Q, Hu S, Du F, et al. Mapping global acceptance and uptake of COVID-19 vaccination: A systematic review and meta-analysis. *Commun Med*. 2022;2(1):1-10. doi:10.1038/s43856-022-00177-6
2. COVID-19 vaccines (dedicated COVID-19 vaccination dashboard). Accessed May 30, 2023. <https://www.who.int/emergencies/diseases/novel-coronavirus-2019/covid-19-vaccines>
3. Mathieu E, Ritchie H, Ortiz-Ospina E, et al. A global database of COVID-19 vaccinations. *Nat Hum Behav*. 2021;5(7):947-953. doi:10.1038/s41562-021-01122-8
4. Coronavirus disease (COVID-19): COVID-19 vaccines and people living with HIV. Accessed July 21, 2022. [https://www.who.int/news-room/questions-and-answers/item/coronavirus-disease-\(covid-19\)-covid-19-vaccines-and-people-living-with-hiv](https://www.who.int/news-room/questions-and-answers/item/coronavirus-disease-(covid-19)-covid-19-vaccines-and-people-living-with-hiv)
5. Davtyan M, Frederick T, Taylor J, Christensen C, Brown BJ, Nguyen AL. Determinants of COVID-19 vaccine acceptability among older adults living with HIV. *Medicine (Baltimore)*. 2022;101(31):e29907. doi:10.1097/MD.00000000000029907
6. Ekstrand ML, Heylen E, Gandhi M, Steward WT, Pereira M, Srinivasan K. COVID-19 Vaccine Hesitancy Among PLWH in South India: Implications for Vaccination Campaigns. *J Acquir Immune Defic Syndr* 1999. 2021;88(5):421-425. doi:10.1097/QAI.0000000000002803
7. Govere-Hwenje S, Jarolimova J, Yan J, et al. Willingness to accept COVID-19 vaccination among people living with HIV in a high HIV prevalence community. *BMC Public Health*. 2022;22(1):1239. doi:10.1186/s12889-022-13623-w
8. Huang X, Yu M, Fu G, et al. Willingness to Receive COVID-19 Vaccination Among People Living With HIV and AIDS in China: Nationwide Cross-sectional Online Survey. *JMIR Public Health Surveill*. 2021;7(10):e31125. doi:10.2196/31125
9. Iliyasu Z, Kwaku AA, Umar AA, et al. Predictors of COVID-19 Vaccine Acceptability among Patients Living with HIV in Northern Nigeria: A Mixed Methods Study. *Curr HIV Res*. 2022;20(1):82-90. doi:10.2174/1570162X19666211217093223
10. Jaiswal J, Krause KD, Martino RJ, et al. SARS-CoV-2 Vaccination Hesitancy and Behaviors in a National Sample of People Living with HIV. *AIDS Patient Care STDs*. 2022;36(1):34-44. doi:10.1089/apc.2021.0144
11. Kabir Sulaiman S, Sale Musa M, Dayyab F, Ismail Tsiga-Ahmed F, Kabir Sulaiman A, Tijjani Bako A. Covid-19 Vaccine Hesitancy Among People Living with HIV (Plhiv) in a Low-Resource Setting: A Multi-Center Study of Prevalence, Correlates and Reasons. Published online October 19, 2022. Accessed October 23, 2022. <https://papers.ssrn.com/abstract=4252042>

12. Lyons N, Bhagwandeem B, Edwards J. Factors Affecting COVID-19 Vaccination Intentions among Patients Attending a Large HIV Treatment Clinic in Trinidad Using Constructs of the Health Belief Model. *Vaccines*. 2023;11(1):4. doi:10.3390/vaccines11010004
13. Mesfin Y, Argaw M, Geze S, Zewdu BT. Factors Associated with Intention to Receive COVID-19 Vaccine Among HIV Positive Patients Attending ART Clinic in Southwest Ethiopia. *Patient Prefer Adherence*. 2021;15:2731-2738. doi:10.2147/PPA.S342801
14. Ortiz-Martínez Y, López-López MÁ, Ruiz-González CE, et al. Willingness to receive COVID-19 vaccination in people living with HIV/AIDS from Latin America. *Int J STD AIDS*. 2022;33(7):652-659. doi:10.1177/09564624221091752
15. Qi L, Yang L, Ge J, Yu L, Li X. COVID-19 Vaccination Behavior of People Living with HIV: The Mediating Role of Perceived Risk and Vaccination Intention. *Vaccines*. 2021;9(11):1288. doi:10.3390/vaccines9111288
16. Shallangwa MM, Musa SS, Iwenya HC, Manirambona E, Iii DELP, Tukur BM. Assessment of COVID-19 vaccine hesitancy among people living with HIV/AIDS: a single-centered study. *PAMJ - One Health*. 2023;10(2). doi:10.11604/pamj-oh.2023.10.2.37945
17. Vallée A, Fourn E, Majerholm C, Touche P, Zucman D. COVID-19 Vaccine Hesitancy among French People Living with HIV. *Vaccines*. 2021;9(4):302. doi:10.3390/vaccines9040302
18. Wu S, Ming F, Xing Z, et al. COVID-19 Vaccination Willingness Among People Living With HIV in Wuhan, China. *Front Public Health*. 2022;10:883453. doi:10.3389/fpubh.2022.883453
19. Liu Y, Han J, Li X, et al. COVID-19 Vaccination in People Living with HIV (PLWH) in China: A Cross Sectional Study of Vaccine Hesitancy, Safety, and Immunogenicity. *Vaccines*. 2021;9(12):1458. doi:10.3390/vaccines9121458
20. Su J, Jia Z, Wang X, et al. Acceptance of COVID-19 vaccination and influencing factors among people living with HIV in Guangxi, China: a cross-sectional survey. *BMC Infect Dis*. 2022;22(1):471. doi:10.1186/s12879-022-07452-w
21. Menza TW, Capizzi J, Zlot AI, Barber M, Bush L. COVID-19 Vaccine Uptake Among People Living with HIV. *AIDS Behav*. 2022;26(7):2224-2228. doi:10.1007/s10461-021-03570-9
22. Al-Yammahi A, Reynolds B, Crawford R, et al. Covid-19 Vaccine Uptake in People Living with HIV. :2.
23. Zhao H, Wang H, Li H, et al. Uptake and adverse reactions of COVID-19 vaccination among people living with HIV in China: a case–control study. *Hum Vaccines Immunother*. 2021;17(12):4964-4970. doi:10.1080/21645515.2021.1991183
24. Munn Z, Stern C, Aromataris E, Lockwood C, Jordan Z. What kind of systematic review should I conduct? A proposed typology and guidance for systematic reviewers in the medical and health

- sciences. *BMC Med Res Methodol*. 2018;18(1):5. doi:10.1186/s12874-017-0468-4
25. Nelson L, Ye H, Schwenn A, Lee S, Arabi S, Hutchins BI. Robustness of evidence reported in preprints during peer review. *Lancet Glob Health*. 2022;10(11):e1684-e1687. doi:10.1016/S2214-109X(22)00368-0
 26. Janda G, Khetpal V, Shi X, Ross JS, Wallach JD. Comparison of Clinical Study Results Reported in medRxiv Preprints vs Peer-reviewed Journal Articles. *JAMA Netw Open*. 2022;5(12):e2245847. doi:10.1001/jamanetworkopen.2022.45847
 27. Africa's vaccine rollout hit by hesitancy – DW – 08/21/2021. dw.com. Accessed June 30, 2023. <https://www.dw.com/en/africa-vaccination-rollout-hindered-by-hesitancy-low-supply/a-58936395>
 28. Ashipala DO, Tomas N, Costa Tenete G. Barriers and Facilitators Affecting the Uptake of COVID-19 Vaccines: A Qualitative Perspective of Frontline Nurses in Namibia. *SAGE Open Nurs*. 2023;9:23779608231158420. doi:10.1177/23779608231158419
 29. Schmidt H, Weintraub R, Williams MA, et al. Equitable allocation of COVID-19 vaccines in the United States. *Nat Med*. 2021;27(7):1298-1307. doi:10.1038/s41591-021-01379-6
 30. Rader B, Astley CM, Sewalk K, et al. Spatial modeling of vaccine deserts as barriers to controlling SARS-CoV-2. *Commun Med*. 2022;2(1):1-11. doi:10.1038/s43856-022-00183-8
 31. Hartonen T, Jermy B, Sõnajalg H, et al. Nationwide health, socio-economic and genetic predictors of COVID-19 vaccination status in Finland. *Nat Hum Behav*. Published online April 20, 2023:1-15. doi:10.1038/s41562-023-01591-z
 32. Nguyen LH, Joshi AD, Drew DA, et al. Self-reported COVID-19 vaccine hesitancy and uptake among participants from different racial and ethnic groups in the United States and United Kingdom. *Nat Commun*. 2022;13(1):636. doi:10.1038/s41467-022-28200-3
 33. Magee LA, Molteni E, Bowyer V, et al. National surveillance data analysis of COVID-19 vaccine uptake in England by women of reproductive age. *Nat Commun*. 2023;14(1):956. doi:10.1038/s41467-023-36125-8
 34. Perry M, Akbari A, Cottrell S, et al. Inequalities in coverage of COVID-19 vaccination: A population register based cross-sectional study in Wales, UK. *Vaccine*. 2021;39(42):6256-6261. doi:10.1016/j.vaccine.2021.09.019
 35. Roederer T, Mollo B, Vincent C, et al. Estimating COVID-19 vaccine uptake and its drivers among migrants, homeless and precariously housed people in France. *Commun Med*. 2023;3(1):1-11. doi:10.1038/s43856-023-00257-1
 36. McGowan VJ, Bambra C. COVID-19 mortality and deprivation: pandemic, syndemic, and endemic health inequalities. *Lancet Public Health*. 2022;7(11):e966-e975. doi:10.1016/S2468-2667(22)00223-7
 37. Schmid P, Rauber D, Betsch C, Lidolt G, Denker ML. Barriers of Influenza Vaccination Intention

- and Behavior – A Systematic Review of Influenza Vaccine Hesitancy, 2005 – 2016. *PLOS ONE*. 2017;12(1):e0170550. doi:10.1371/journal.pone.0170550
38. Wijayanti KE, Schütze H, MacPhail C, Braunack-Mayer A. Parents' knowledge, beliefs, acceptance and uptake of the HPV vaccine in members of The Association of Southeast Asian Nations (ASEAN): A systematic review of quantitative and qualitative studies. *Vaccine*. 2021;39(17):2335-2343. doi:10.1016/j.vaccine.2021.03.049
39. Garrett PM, White JP, Lewandowsky S, et al. The acceptability and uptake of smartphone tracking for COVID-19 in Australia. *PLOS ONE*. 2021;16(1):e0244827. doi:10.1371/journal.pone.0244827
40. Aluisio AR, Lim RK, Tang OY, et al. Acceptability and uptake of HIV self-testing in emergency care settings: A systematic review and meta-analysis. *Acad Emerg Med Off J Soc Acad Emerg Med*. 2022;29(1):95-104. doi:10.1111/acem.14323
41. Nadarzynski T, Frost M, Miller D, et al. Vaccine acceptability, uptake and completion amongst men who have sex with men: A systematic review, meta-analysis and theoretical framework. *Vaccine*. 2021;39(27):3565-3581. doi:10.1016/j.vaccine.2021.05.013
42. Ropka ME, Keim J, Philbrick JT. Patient Decisions About Breast Cancer Chemoprevention: A Systematic Review and Meta-Analysis. *J Clin Oncol*. 2010;28(18):3090-3095. doi:10.1200/JCO.2009.27.8077
43. World Health Organization. *Methodology for the Assessment of Missed Opportunities for Vaccination*. World Health Organization; 2017. Accessed April 29, 2022. <https://apps.who.int/iris/handle/10665/259201>
44. Ssentongo P, Heilbrunn ES, Ssentongo AE, et al. Epidemiology and outcomes of COVID-19 in HIV-infected individuals: a systematic review and meta-analysis. *Sci Rep*. 2021;11(1):6283. doi:10.1038/s41598-021-85359-3
45. Jakharia N, Subramanian AK, Shapiro AE. COVID-19 in the Immunocompromised Host, Including People with Human Immunodeficiency Virus. *Infect Dis Clin*. 2022;36(2):397-421. doi:10.1016/j.idc.2022.01.006
46. Ye Y, Zhang Q, Wei X, Cao Z, Yuan HY, Zeng DD. Equitable access to COVID-19 vaccines makes a life-saving difference to all countries. *Nat Hum Behav*. 2022;6(2):207-216. doi:10.1038/s41562-022-01289-8
47. Chen L, Xu F, Han Z, et al. Strategic COVID-19 vaccine distribution can simultaneously elevate social utility and equity. *Nat Hum Behav*. 2022;6(11):1503-1514. doi:10.1038/s41562-022-01429-0
48. Moore S, Hill EM, Dyson L, Tildesley MJ, Keeling MJ. Retrospectively modeling the effects of increased global vaccine sharing on the COVID-19 pandemic. *Nat Med*. 2022;28(11):2416-2423. doi:10.1038/s41591-022-02064-y

49. Nehal KR, Steendam LM, Campos Ponce M, van der Hoeven M, Smit GSA. Worldwide Vaccination Willingness for COVID-19: A Systematic Review and Meta-Analysis. *Vaccines*. 2021;9(10):1071. doi:10.3390/vaccines9101071
50. Wang Q, Yang L, Jin H, Lin L. Vaccination against COVID-19: A systematic review and meta-analysis of acceptability and its predictors. *Prev Med*. 2021;150:106694. doi:10.1016/j.ypmed.2021.106694
51. Kazemina M, Afshar ZM, Rajati M, Saeedi A, Rajati F. Evaluation of the Acceptance Rate of Covid-19 Vaccine and its Associated Factors: A Systematic Review and Meta-analysis. *J Prev*. 2022;43(4):421-467. doi:10.1007/s10935-022-00684-1
52. Robinson E, Jones A, Lesser I, Daly M. International estimates of intended uptake and refusal of COVID-19 vaccines: A rapid systematic review and meta-analysis of large nationally representative samples. *Vaccine*. 2021;39(15):2024-2034. doi:10.1016/j.vaccine.2021.02.005

Decision Letter, first revision:

4th September 2023

Dear Dr. Sulaiman,

Thank you for your patience as we've prepared the guidelines for final submission of your Nature Human Behaviour manuscript, "The global prevalence and determinants of COVID-19 vaccine acceptance and uptake in people living with HIV: a systematic review and meta-analyses" (NATHUMBEHAV-23030675A). Please carefully follow the step-by-step instructions provided in the attached file, and add a response in each row of the table to indicate the changes that you have made. Please also check and comment on any additional marked-up edits we have proposed within the text. Ensuring that each point is addressed will help to ensure that your revised manuscript can be swiftly handed over to our production team.

We would hope to receive your revised paper, with all of the requested files and forms within two-three weeks. Please get in contact with us if you anticipate delays.

If you have not done so already, please alert us to any related manuscripts from your group that are under consideration or in press at other journals, or are being written up for submission to other journals (see:

<https://www.nature.com/nature-research/editorial-policies/plagiarism#policy-on-duplicate-publication> for details).

Nature Human Behaviour offers a Transparent Peer Review option for new original research manuscripts submitted after December 1st, 2019. As part of this initiative, we encourage our authors to support increased transparency into the peer review process by agreeing to have the reviewer comments, author rebuttal letters, and editorial decision letters published as a Supplementary item. When you submit your final files please clearly state in your cover letter whether or not you would like to participate in this initiative. Please note that failure to state your preference will result in delays in accepting your manuscript for publication.

In recognition of the time and expertise our reviewers provide to Nature Human Behaviour's editorial process, we would like to formally acknowledge their contribution to the external peer review of your manuscript entitled "The global prevalence and determinants of COVID-19 vaccine acceptance and uptake in people living with HIV: a systematic review and meta-analyses". For those reviewers who give their assent, we will be publishing their names alongside the published article.

Cover suggestions

We welcome submissions of artwork for consideration for our cover. For more information, please see our https://www.nature.com/documents/Nature_covers_author_guide.pdf target="new"> guide for cover artwork.

ORCID

Non-corresponding authors do not have to link their ORCIDs but are encouraged to do so. Please note that it will not be possible to add/modify ORCIDs at proof. Thus, please let your co-authors know that if they wish to have their ORCID added to the paper they must follow the procedure described in the following link prior to acceptance:

Nature Human Behaviour has now transitioned to a unified Rights Collection system which will allow our

Author Services team to quickly and easily collect the rights and permissions required to publish your work. Approximately 10 days after your paper is formally accepted, you will receive an email in providing you with a link to complete the grant of rights. If your paper is eligible for Open Access, our Author Services team will also be in touch regarding any additional information that may be required to arrange payment for your article.

Please note that *Nature Human Behaviour* is a Transformative Journal (TJ). Authors may publish their research with us through the traditional subscription access route or make their paper immediately open access through payment of an article-processing charge (APC). Authors will not be required to make a final decision about access to their article until it has been accepted. Find out more about Transformative Journals

Please use the following link for uploading these materials:
[REDACTED]

Best regards,
Rebecca Ireton
Editorial Assistant
Nature Human Behaviour

On behalf of

Arunas Radzvilavicius, PhD
Senior Editor, Nature Human Behaviour

Nature Research

Reviewer #1:

Remarks to the Author:

After reviewing the revised documents, I am pleased to inform you that the feedback is satisfactory. I have no further comments to add at this time.

Reviewer #2:

Remarks to the Author:

The authors have responded to all of my comments.

Additional corrections are needed in several places of the added tables and text. Table 4 and 5 correctly report the first outcome as “gender”, however, the labels used incorrectly denote sex. Male (vs. female) should be changed to Men (vs. women). Similarly, the text on pages 24, 25, 28/29 (and anywhere else) should be changed to men vs. woman.

p. 29. “blacks (vs. others)” should be changed to Black people. One also wonders since only 2 studies are reported in the meta-analysis on race/ethnicity, if these 2 studies reflect all other race/ethnicities as the referent.

Author Rebuttal, first revision:

REFEREE 2

Comment (1)

Remarks to the Author:

The authors have responded to all of my comments.

Additional corrections are needed in several places of the added tables and text. Table 4 and 5 correctly report the first outcome as “gender”, however, the labels used incorrectly denote sex.

Male (vs. female) should be changed to Men (vs. women). Similarly, the text on pages 24, 25, 28/29 (and anywhere else) should be changed to men vs. woman.

p. 29. “blacks (vs. others)” should be changed to Black people. One also wonders since only 2 studies are reported in the meta-analysis on race/ethnicity, if these 2 studies reflect all other race/ethnicities as the referent.

Response

We thank the Reviewer for these important observations. We have replaced Male with Men and Female with Women” in all appropriate sections of the manuscript as you suggested. We have re-analyzed the data and classified the race variable Black people vs. white, and people of other races (Latinx/Hispanic/Mixed race) vs. White. Interestingly, this were statistically significant. Please see the table 3 and four below.

Table 3: Results of meta-analyses of determinants of COVID-19 vaccine acceptance

Outcome	Studies (N) and references	WHO region(s) represented	OR (95% CI)	p-Value	I ² Within
Gender (male vs female) (men vs women)	14 16 ^{60-62,64,69,70,72,75-78,80,90,94,95,108}	AFR, AMR, EUR, SEAR, WPR	2.29 (1.19 - 4.39) 2.06 (1.16 - 3.66)	0.01	95%
Age (< 40 years vs >= 40 years)	8-9 ^{33,60,62,70,75-77,80,95}	AFR, AMR, EUR, WPR	0.70 (0.52 - 0.93) 0.70 (0.54 - 0.90)	0.01 0.006	70%
Marital status (single/divorced/widowed vs married/cohabited)	10 13 ^{33,60,61,69,70,75,78,80,90,94,95,98,108}	AFR, AMR, WPR	1.02 (0.86 - 1.21) 0.96 (0.80 - 1.15)	0.83	38%

Race (Black vs others) (Black vs White)	2 ^{72,78}	AMR	0.75 (0.41 - 1.35) 0.50 (0.27 - 0.94)	0.34 0.03	75%
Race (Others [Latinx/Hispanic/Mixed race] vs White)	2 ^{72,78}	AMR	1.04 (0.49 - 2.20)	0.92	
Education (primary and below vs secondary and above)	11 14 ^{33,57,60,61,64,69,70,72,75,80,90, 95,98,108}	AFR, AMR, SEAR, WPR	0.69 (0.43 - 1.10) 0.60 (0.40 - 0.89)	0.12 0.01	93%
Employment status (unemployed vs employed)	9 11 ^{33,69,70,72,75,78,80,90,94,95,108}	AFR, AMR, WPR	0.80 (0.57 - 1.13) 0.96 (0.66 - 1.31)	0.20	77%
Income (low vs medium/high)	7 9 ^{60,61,69,70,75,78,80,95,98}	AFR, AMR, WPR	0.96 (0.69 - 1.35) 0.96 (0.74 - 1.24)	0.83 0.74	75%
Residence (Rural vs urban)	4 ^{77,80,95,108}	AFR, AMR	1.69 (1.33 - 2.14)	<0.00001	94%
Comorbidity (absent vs present)	8 9 ^{60-62,69,70,75,77,90,98}	AFR, AMR, EUR, WPR	0.72 (0.42 - 1.25) 0.81 (0.49 - 1.34)	0.24 0.41	91%
Vaccine safety concern (yes vs no)	4 ^{57,62,64,70}	AFR, EUR, SEAR, WPR	0.41 (0.32 - 0.53)	<0.00001	96%
Perceived vaccine effectiveness (yes vs no)	3 ^{57,62,70}	AFR, EUR, WPR	1.80 (1.27 - 2.56)	0.001	93%
Vaccine trust (yes vs no)	3 ^{62,64,90}	AMR, EUR, SEAR	15.17 (9.16 - 25.12)	<0.00001	31%
Perceived susceptibility to COVID-19 (yes vs no)	3 ^{61,62,80}	AFR, EUR, WPR	1.34 (1.07 - 1.68)	0.01	0%
Fear of COVID-19 effect on PLHIV (yes vs no)	4 ^{61,62,70,76}	AFR, EUR, WPR	2.01 (1.60 - 2.54)	<0.00001	89%

Know someone who died of COVID-19 (yes vs no)	2 ^{62,64}	EUR, SEAR	1.06 (0.68 - 1.66)	0.78	82%
Recent of influenza vaccination (yes vs no)	4 ^{62,75-77}	AFR, AMR, EUR, WPR	1.53 (1.29 - 1.81)	<0.00001	36%

Bold font within the p-Value column indicates statistical significance

Table 4: Results of meta-analyses of determinants of COVID-19 vaccine uptake

Outcome	Studies (N) and references	WHO region(s) represented	OR (95% CI)	p-Value	I ² Within
Gender (male vs female) (men vs women)	5 ^{60,63,67,72,74} 6 ^{60,63,67,72,74,106}	AMR, EUR, WPR	1.56 (1.38 - 1.77) 1.55 (1.27 - 1.89)	<0.00001	33%
Age (<40 years vs >= 40 years)	3 ^{60,63,74} 4 ^{60,63,74,106}	AMR, WPR	0.59 (0.54 - 0.65) 0.58 (0.53 - 0.64)	<0.00001	96%
Marriage/cohabitation (single/divorced/widowed vs married/cohabited)	2 ^{60,74}	WPR	0.86 (0.68 - 1.09)	0.21	68%
Race (Black vs others)	3 ^{63,67,72}	AMR, EUR	0.67 (0.57 - 0.77)	<0.00001	0%
Race (Black vs White)	3 ^{63,72,106}	AMR	0.60 (0.52 - 0.70)	<0.00001	66%

Race (Others [Latinx/Hispanic/Mixed race] vs White)	2 ^{63,72}	AMR	0.31 (0.28 - 0.34)	<0.00001	91%
Education (primary and below vs secondary and above)	3 ^{60,72,74} 5 ^{60,72,74,98,106}	AMR, WPR	0.49 (0.38 - 0.62) 0.50 (0.41 - 0.61)	<0.00001	73%
Income (low vs medium/high)	2 ^{60,106}	AMR, WPR	0.91 (0.61 - 1.34)	0.62	34%
Employment (unemployed vs employed)	3 ^{67,72,74} 4 ^{67,72,74,106}	AMR, EUR, WPR	0.63 (0.47 - 0.84) 0.56 (0.43 - 0.73)	0.002	74%
Receipt of influenza vaccination (yes vs no)	2 ^{63,74}	AMR, WPR	6.73 (6.11 - 7.14)	<0.00001	93%

Bold font within the p-Value column indicates statistical significance

Final Decision Letter:

Dear Dr Sulaiman,

We are pleased to inform you that your Article "A systematic review and meta-analyses of the global prevalence and determinants of COVID-19 vaccine acceptance and uptake in people living with HIV", has now been accepted for publication in Nature Human Behaviour.

Please note that *Nature Human Behaviour* is a Transformative Journal (TJ). Authors may publish their research with us through the traditional subscription access route or make their paper immediately open access through payment of an article-processing charge (APC). Authors will not be required to make a final decision about access to their article until it has been accepted. Find out more about Transformative Journals

Authors may need to take specific actions to achieve compliance with funder and institutional open access mandates. If your research is supported by a funder that requires immediate open access (e.g. according to Plan S principles) then you should select the gold OA route, and we will direct you to the compliant route where possible. For authors selecting the subscription publication route, the journal's standard licensing terms will need to be accepted, including self-archiving policies. Those licensing terms

will supersede any other terms that the author or any third party may assert apply to any version of the manuscript.

With best regards,

Arunas Radzvilavicius, PhD
Senior Editor, Nature Human Behaviour
Nature Research